# Stakeholder integration predicts better outcomes from groundwater sustainability policy

Debra Perrone [1,16] ✉, Melissa M. Rohde [2,3,4,16] ✉,
Courtney Hammond Wagner [5,6,16] ✉, Rebecca Anderson[7,8], Samantha Arthur[9],
Ngodoo Atume[10], Meagan Brown[11], Lauren Esaki-Kua[6],
Martha Gonzalez Fernandez[1], Kelly A. Garvey[6,11], Katherine Heidel [12],
William D. Jones[1,5], Sara Khosrowshahi Asl[6,13], Carrie Munill[12], Rebecca Nelson[14],
J. Pablo Ortiz-Partida [15] & E. J. Remson[2]

Natural resources policies that promote sustainable management are critical for protecting diverse stakeholders against depletion. Although integrating diverse stakeholders into these policies has been theorized to improve protection, empirical evidence is lacking. Here, we evaluate 108 Sustainability Plans under California's Sustainable Groundwater Management Act to quantify how well stakeholders are integrated into plans and protected from groundwater depletion. We find that the majority of Sustainability Plans do not integrate or protect the majority of their stakeholders. Nevertheless, our results show that when stakeholders are more integrated into a Sustainability Plan, they are more likely to be protected, particularly for those that lack formal access to decision-making processes. Our findings provide strong empirical evidence that integrating diverse stakeholders into sustainability planning is beneficial for stakeholders who are vulnerable to the impacts of natural resource depletion.

Groundwater is an essential resource for supporting sustainable food systems, healthy communities, and ecosystems. Nevertheless, groundwater depletion is becoming one of the most prominent natural resource challenges facing society[1,2], with thousands of researchers and practitioners calling for more sustainable management[3,4]. In theory, sustainable groundwater management ensures that current and future societal, ecological, and economic needs of all user groups are met or protected[5], but in practice, some user-groups' needs may be overlooked. Globally, the 21st century has seen nations and sub-national units moving away from unmanaged natural resources through the development of policies to guide and constrain resource use[6]. The exact approaches to management are as varied as their outcomes[7], but the incorporation of stakeholders, their knowledge, and needs − herein stakeholder integration − into natural resource policy processes has been posited to result in better outcomes among user groups[8–11]. In fact, natural resource policies around the globe are

[1]Environmental Studies, University of California Santa Barbara, Santa Barbara, CA, USA. [2]California Water Program, The Nature Conservancy, Sacramento, CA, USA. [3]SUNY College of Environmental Science and Forestry, Syracuse, NY, USA. [4]Rohde Environmental Consulting, LLC, Seattle, WA, USA. [5]USDA Agricultural Research Service, Food Systems Research Unit, Burlington, VT, USA. [6]Water in the West, Stanford University, Stanford, CA, USA. [7]Independent Consultant, Portland, OR, USA. [8]WaterNow Alliance, San Francisco, CA, USA. [9]Audubon California, Sacramento, CA, USA. [10]Clean Water Action, Oakland, CA, USA. [11]Bren School of Environmental Science and Management, University of California Santa Barbara, Santa Barbara, CA, USA. [12]Tetra Tech, Lafayette, CA, USA. [13]Department of Environmental Science, Policy, and Management, University of California, Berkeley, CA, USA. [14]Melbourne Law School, University of Melbourne, Melbourne, VIC, Australia. [15]Union of Concerned Scientists, Oakland, CA, USA. [16]These authors contributed equally: Debra Perrone, Melissa M. Rohde, Courtney Hammond Wagner. ✉e-mail: perrone@ucsb.edu; melissa@rohdeenvironmental.com; courtney.hammond-wagner@usda.gov

increasingly inviting local stakeholders who may not have formal governance authority to participate in policy processes, a trend reflected in groundwater management[7,8,11]. Nevertheless, there is little empirical evidence evaluating the impact of stakeholder integration on natural resource outcomes[8,12,13]. Consequently, we do not know if stakeholder integration actually leads to policies that protect stakeholders[6,14].

California's Sustainable Groundwater Management Act provides an unprecedented opportunity[15] to assess whether stakeholder integration into planning is associated with stakeholder protection (Supplementary Section 1). The Sustainable Groundwater Management Act takes a decentralized approach to groundwater management, mandating newly-designated local Groundwater Sustainability Agencies[16] to achieve sustainability within 20 years through the development and implementation of Groundwater Sustainability Plans (Sustainability Plans). This proliferation of local policy processes has resulted in more than 100 Sustainability Plans that detail stakeholders and set management thresholds quantifying undesirable results linked to depletion. These Sustainability Plans operate within the same general legal and governance context for the same resource system. Collectively, the Sustainability Plans provide a large dataset with which to examine the

association between stakeholder integration and protection from depletion across diverse user-groups, geographical contexts, and biogeophysical dynamics.

The objective of this paper is to assess if greater stakeholder integration into Sustainability Plans leads to better outcomes for those stakeholders. We focus on three stakeholder groups: agriculture, domestic, and environment (Fig. 1a). These three groups represent groups of individual users that self-supply groundwater for diverse uses, such as food production, household supply, and ecosystem function. We define integration as how well each Sustainability Plan incorporates stakeholders, their knowledge, and needs across four components: *engage*, *describe*, *analyze*, and *act* (Fig. 1b). We use the four components as a proxy for stakeholder integration into the local policy processes that resulted in each of the Sustainability Plans; the plans reflect only the stated and visible priorities of the Groundwater Sustainability Agencies, and not invisible power dynamics that determine decision-making powers within these local agencies. We assess outcomes by quantifying how many stakeholders are protected from losing access to groundwater based on quantified thresholds defined within each Sustainability Plan (Fig. 1c, d).

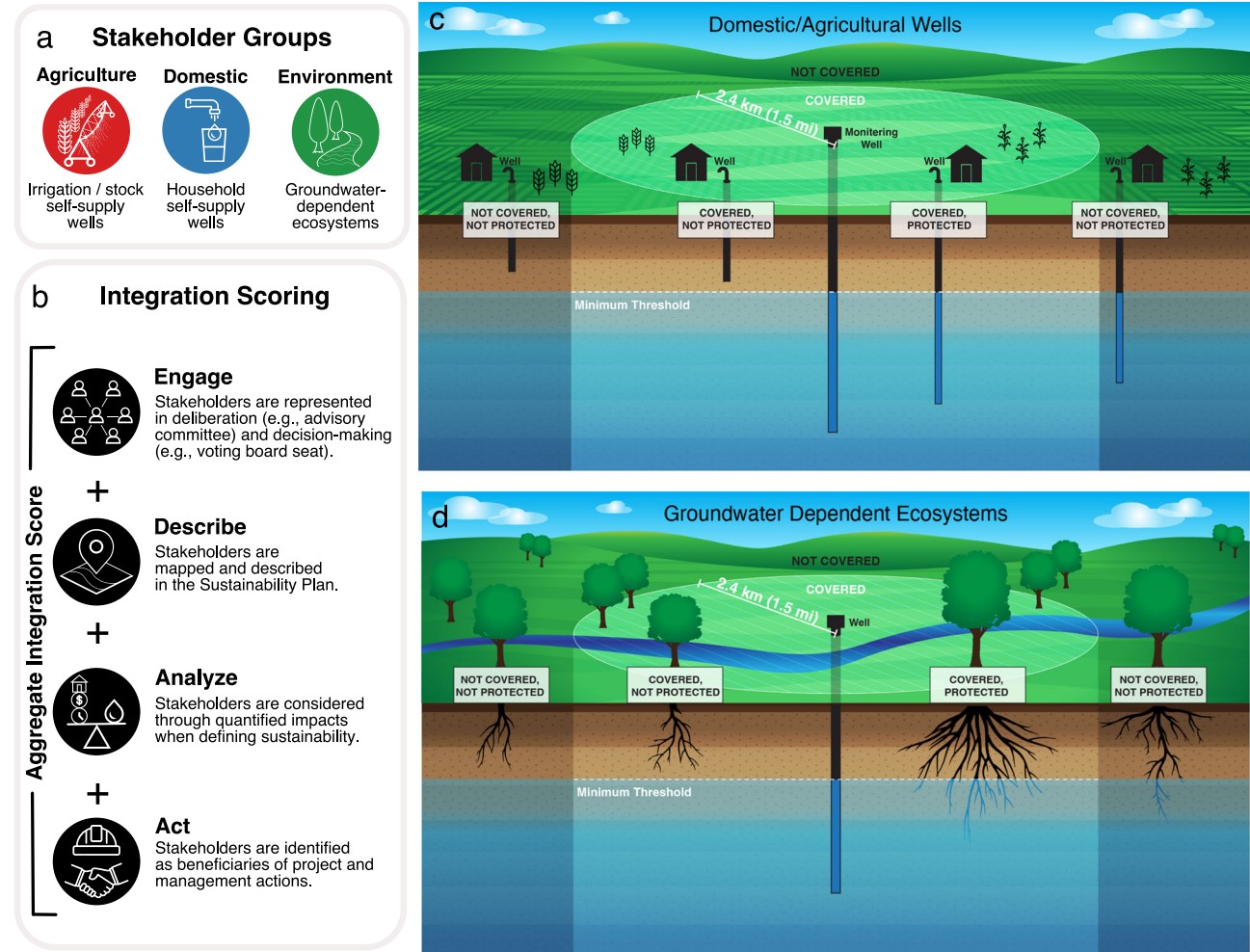

**Fig. 1 | Conceptual framework for evaluating Sustainability Plans and spatially assessing stakeholder protection. a** Stakeholder groups and their definitions. **b** Stakeholder integration scoring components and their definitions. **c, d** Spatial assessment: for each instance of a stakeholder ((**c**) well or (**d**) ecosystem), we identified monitoring wells that were at or within a horizontal distance of -2.4 km (1.5 miles). Stakeholders nearby monitoring wells were considered covered and were assessed for how well the Sustainability Plan protects the stakeholder from losing access to water as a result of declining groundwater levels. Individual stakeholders not covered by a monitoring well were deemed not protected. For covered wells, protection was determined by comparing the depth of each stakeholder to the minimum threshold established at the nearby monitoring well. If the depth of a stakeholder was equal to or shallower than the minimum threshold at the nearby monitoring well, the stakeholder was not protected.

To meet our objective, we use a mixed-methods approach by (1) developing and implementing a quantitative assessment of stakeholder integration into 108 Sustainability Plans based on theoretical scholarship on stakeholder integration; (2) conducting geospatial analyses of stakeholder protection from declining groundwater levels; and (3) combining the results of the first two approaches and applying multiple statistical methods to assess if greater stakeholder integration is associated with more protection. (4) In addition, we extend the aforementioned analyses to small farms[17] and disadvantaged communities (CA Water Code 79505.5) − subgroups that are considered economically vulnerable to groundwater depletion because of the high costs[18] required to adapt to declining water levels − when the data are available.

Here, we show that: (1) 91% of Sustainability Plans did not integrate stakeholders comprehensively and (2) Sustainability Plans do not protect 60% of agricultural wells, 63% of domestic wells, and 91% of ecosystems. Nevertheless, our assessment suggests that (3) when Sustainability Plans integrate a stakeholder group, those stakeholders are more likely to be protected, particularly for the domestic and environment groups. (4) Additionally, our results show that subgroups that are considered more economically vulnerable to groundwater depletion are less integrated than their respective broader stakeholder groups; where data permitted analyses, our results show that they are also less protected. Our findings highlight a greater need for stakeholder integration into the development and implementation of Sustainability Plans to ensure better outcomes among diverse stakeholders. Because lessons on governance and sustainability from groundwater can inform other natural resources management challenges, our results suggest that sustainability is more likely to be achieved across diverse user-groups if these stakeholders are integrated into policy processes.

## Results

### Incomplete and variable stakeholder integration

We evaluated 108 Sustainability Plans through reading and characterizing 162,943 pages of text. We developed a rubric to assess the degree to which the agriculture, domestic, and environment groups (Supplementary Section 2; Supplementary Fig. 2.1; Supplementary Tables 2.1–2.3) were explicitly mentioned within each Sustainability Plan. More specifically, we defined four generalizable integration components (Fig. 1b), for which we asked, how well did each Sustainability Plan: *engage, describe, analyze* impacts of depletion on, and *act* to support each stakeholder group and subgroup as evidenced within the text? Our rubric enabled us to calculate scores (ranging from zero to two) for each integration component, and to sum integration components to make an *aggregate* score (ranging from zero to eight); the *aggregate* score assessed how well each Sustainability Plan integrated each stakeholder group across the four components (Supplementary Fig. 2.3). Most Sustainability Plans failed to comprehensively integrate stakeholders: only 9% of plans achieved a score >0 for all components for all stakeholder groups. In particular, *engage* and *analyze* had the lowest scores for the majority of stakeholder groups (Fig. 2). Our results suggest that the four components do not progress sequentially, and thus, do not represent coordinated components of stakeholder integration.

We statistically compared integration components and *aggregate* scores between stakeholder groups to examine if there were significant differences in each group's integration into the Sustainability Plans (Supplementary Table 2.4; Supplementary Fig. 2.4). Comparing agricultural, domestic, and environmental stakeholders via Kruskal–Wallis tests and pairwise Dunn tests, we saw no overall pattern between stakeholder integration component scores − no stakeholder group scored consistently higher or lower than other groups across components − despite significant ($p \le 0.01$) differences existing for specific components between groups. Agriculture scored higher for

the *engage* ($p \le 0.001$) and *act* ($p \le 0.001$) components but lower for *describe* ($p \le 0.01$) and *analyze* ($p \le 0.01$) than domestic. The domestic group scored lower for *engage* ($p \le 0.01$), *describe* ($p \le 0.001$), and *act* ($p \le 0.001$) but higher for *analyze* ($p \le 0.001$) than the environment group. On the *aggregate* score, the environment group scored higher than the agriculture group ($p \le 0.05$) and domestic group ($p \le 0.01$). The percentage of Sustainability Plans with high *aggregate* scores (scores 5–8) varied only slightly among groups: 31% for agriculture, 31% for domestic, and 39% for environment. Environment's slightly higher *aggregate* score was driven by an exceptionally high score for *describe*. In general, we find no compelling evidence in our analysis that any stakeholder group was more integrated than another.

We ran a sensitivity analysis with alternative definitions of *engage* for agriculture − definitions which accounted for the group's power to form a Groundwater Sustainability Agency via a governing entity that provides water primarily for agricultural use, e.g., reclamation or irrigation district, which is not an option available to the other stakeholder groups (Supplementary Section 3). Irrigation and reclamation districts are governing entities that in practice serve primarily agricultural interests and can act as Groundwater Sustainability Agencies. We anticipate that this method of accounting for agricultural representation is a conservative estimate, as it is likely that agricultural interests are represented by other entities serving as or on Groundwater Sustainability Agency boards that were not included here. Using these alternative definitions, the agriculture group scored higher for the *engage* component than the domestic and environment groups, and agriculture's *aggregate* integration score was on par with the environment's score. In short, our overall interpretation of integration remains similar to those found using the definition of *engage* used for the domestic and environment groups (Supplementary Table 3.3).

### Most wells and ecosystems are not protected

We analyzed coverage and protection by stakeholder group using each Sustainability Plan's groundwater level minimum thresholds established at representative monitoring wells (Fig. 3; Supplementary Sections 4, 5). Minimum thresholds measure where undesirable results (e.g., a well running dry or an ecosystem die-off) may occur if groundwater levels decline. For each instance of a stakeholder (i.e., domestic well, agricultural well, or groundwater-dependent ecosystem), we identified monitoring wells that were nearby: at or within a horizontal distance of ~2.4 km (1.5 miles; Fig. 1c, d). Stakeholders nearby monitoring wells were considered covered and were assessed for how well the Sustainability Plan protected the stakeholder from losing access to water as a result of declining groundwater levels. Individual stakeholders not covered by a monitoring well were deemed not protected. For wells that are covered, protection was determined by comparing each stakeholder's groundwater access depth (e.g., well depth or maximum rooting depth for vegetation) to the minimum threshold groundwater level established at the nearby monitoring well. If a stakeholder's groundwater access depth was equal to or shallower than the minimum threshold at the nearby monitoring well, the stakeholder was not protected (Fig. 1c, d).

Across all Sustainability Plans, 49% of agricultural wells ($n_{covered} = 18,520$), 49% of domestic wells ($n_{covered} = 42,716$), and 42% of ecosystems ($n_{covered} = 645$ km²) were covered by nearby monitoring wells (Supplementary Table 4.1). Subsequently, 40% of agricultural wells ($n_{protected} = 14,964$), 37% of domestic wells ($n_{protected} = 32,449$), and 9% of ecosystems ($n_{protected} = 138$ km²) were protected by the Sustainability Plans. Using the Pearson's chi-squared test, we found no significant difference in coverage between agricultural and domestic wells ($p > 0.05$), but found domestic wells to be significantly ($p \le 0.001$) less protected than agricultural wells (Supplementary Table 4.2). Coverage and protection for ecosystems were significantly lower than for domestic wells and agricultural wells ($p \le 0.001$ for all comparisons).

We ran a sensitivity analysis, adjusting the horizontal distance surrounding each monitoring well – ~0.8 km to ~4.0 km – to assess the impact on coverage and protection (Supplementary Table 4.1–4.4, Supplementary Figs. 4.1–4.7). As the horizontal distance increased, the proportion of wells and ecosystems covered and protected within each Plan increased. The increases in protection did not scale proportionally across stakeholders. For example, using the largest horizontal distance (~4.0 km), 78%, 77%, and 61% of agricultural wells,

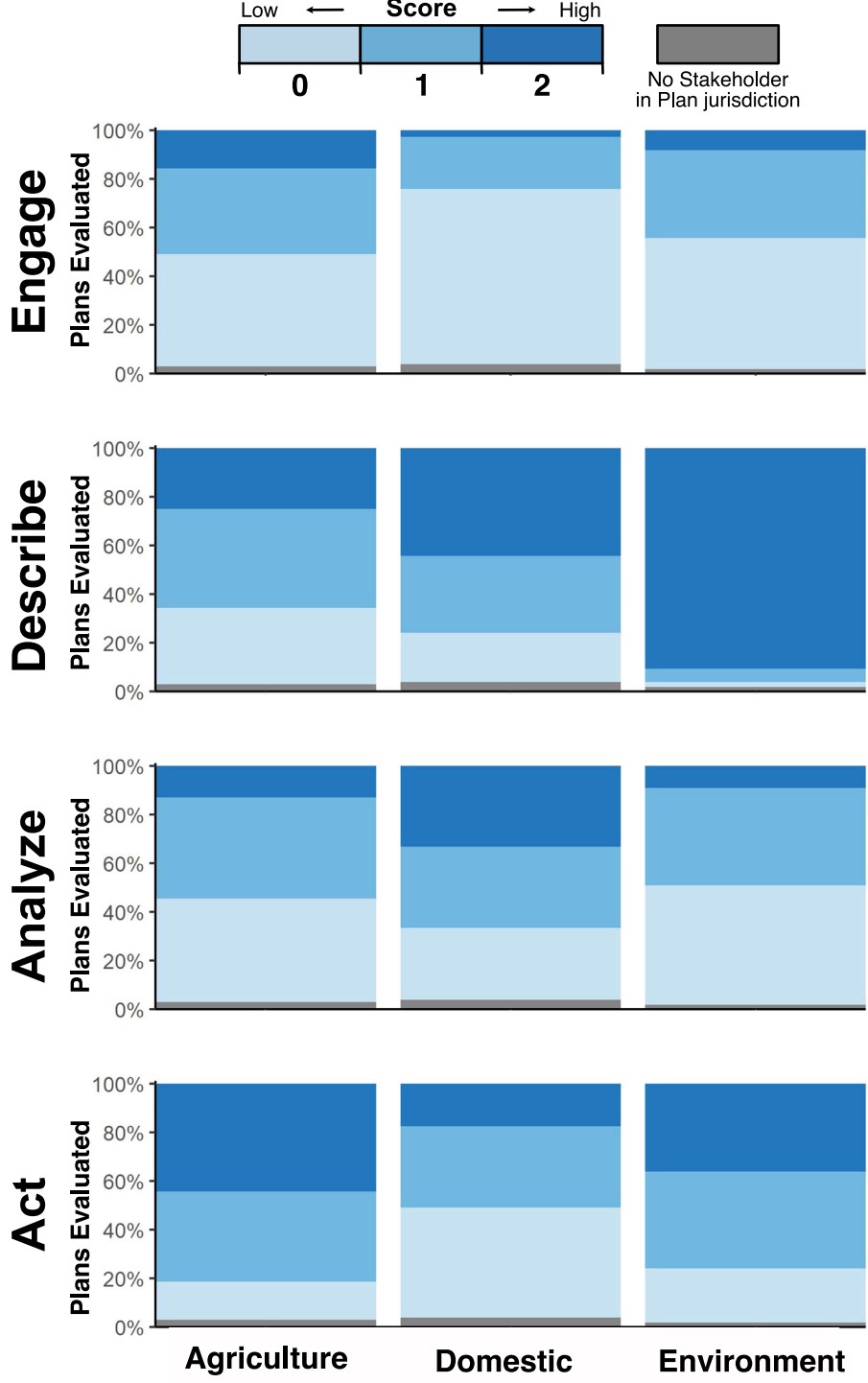

**Fig. 2 | Stakeholder integration scores for stakeholder groups (agriculture, domestic, environment).** Integration scores range between zero and two for each integration component: *engage, describe, analyze,* and *act*. For each stakeholder group, subplots show the percentage of Sustainability Plans evaluated that received a zero, one, or two score within each integration component. Sustainability Plans without a stakeholder group are depicted in gray. The majority of Sustainability Plans (91%) did not comprehensively integrate all three stakeholder groups, which we defined as: addressing each of the four stakeholder integration components (i.e., score >0) across the three stakeholder groups. Comparing the integration components (top to bottom) across the three stakeholder groups (left to right), we found no clear patterns: the agriculture and environment groups scored higher on *engage* than the domestic group; the environment group scored higher on *describe* than the domestic and agriculture groups; the domestic group scored higher on *analyze* than the agriculture and environment groups; and the agriculture and environment groups score higher on *act* than the domestic group.

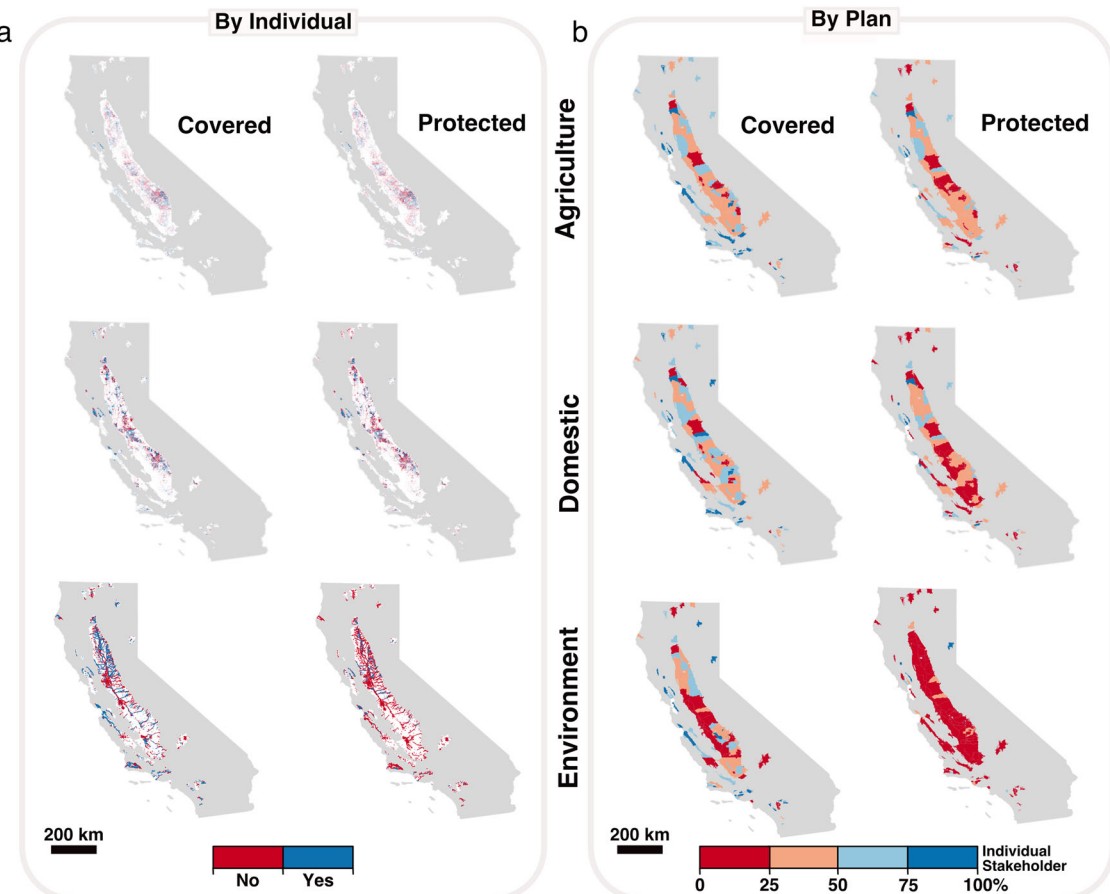

**Fig. 3 | Coverage and protection of each stakeholder group by groundwater level minimum thresholds established at representative monitoring wells.**
**a** Coverage and protection shown for each agricultural well (top), domestic well (middle) and groundwater-dependent ecosystem (bottom). **b** Coverage and protection for each stakeholder summarized within the Sustainability Plan area. The proportion of Sustainability Plans with ≥50% covered agricultural wells and domestic wells within a given plan was 64% ($n$ = 67 of 105 Sustainability Plans) and

64% ($n$ = 66 of 104 Sustainability Plans), respectively (depicted by light or dark blue). The proportion of Sustainability Plans with ≥50% protected agricultural wells and domestic wells within a given plan was 37% ($n$ = 39 of 105 Sustainability Plans) and 29% ($n$ = 30 of 104 Sustainability Plans), respectively. The proportion of Sustainability Plans with ≥50% covered and protected groundwater-dependent ecosystems within a given plan was 54% ($n$ = 57 out of 106 Sustainability Plans) and 8% ($n$ = 8 out of 106 Sustainability Plans), respectively.

domestic wells, and ecosystems were covered, but only 64%, 59%, and 15% were protected, respectively. In short, the disparity in protection among agricultural, domestic, and environmental stakeholders was exacerbated as we increased the horizontal distance used to assess coverage.

In addition to quantifying protection, we performed a review of each Sustainability Plan's "significant and unreasonable" conditions to examine if and how these are established to protect stakeholders (Cal. Water Code § 10721(x), 23 Cal. Code of Regs. §§ 354.26, 354.28). Groundwater Sustainability Agencies are given the discretion to define minimum thresholds in order to avoid "significant and unreasonable" conditions. Twenty-four percent of Sustainability Plans ($n$ = 26) established minimum thresholds specifically to be protective of agricultural wells, domestic wells, groundwater-dependent ecosystems, or a combination of stakeholders (Supplementary Tables 4.5–4.6). The remaining 76% used non-stakeholder related approaches to define minimum thresholds, such as lowest historical well levels.

### Stakeholder integration associated with protection

We quantified stakeholder integration as the *aggregate* score for each stakeholder group. To test if greater stakeholder integration into a Sustainability Plan was associated with increased protection for each stakeholder group, we divided each group's *aggregate* integration scores into tertiles (i.e., low integration, moderate integration, and

high integration for each stakeholder group). Then, we used a t-test to compare the mean protection values of plans with low integration and plans with high integration for each stakeholder. Protection was significantly greater for plans with high integration scores than for plans with low integration scores for domestic ($\bar{x}_{low}$ = 25.89% vs $\bar{x}_{high}$ = 46.59%, $t$ = 3.68, $p \leq 0.001$) and environmental ($\bar{x}_{low}$ = 6.25% vs $\bar{x}_{high}$ = 18.67%, $t$ = 2.80, $p \leq 0.01$) stakeholders, but not for agricultural stakeholders ($\bar{x}_{low}$ = 40.81% vs $\bar{x}_{high}$ = 48.23%, $t$ = 1.28, $p > 0.05$; Supplementary Table 6.1). In short, greater integration into Sustainability Plans is associated with greater protection for the domestic and environment groups.

To assess if integration is associated with more similar protection values among stakeholders, we performed two ANOVAs – one among stakeholders' mean protection values associated with low integration scores (i.e., first tertile) and one among stakeholders' mean protection values associated with high integration scores (i.e., third tertile) – followed by Tukey's pairwise comparisons (Fig. 4). For Sustainability Plans with low integration scores, we found significant differences among protection values ($F$ = 27.84, $p \leq 0.001$; Supplementary Table 6.2). Moreover, the mean protection score for the domestic group was significantly lower than for the agriculture group ($p \leq 0.01$), and the mean protection score for the environment group was significantly lower than for the agriculture and domestic groups ($p \leq 0.001$). For Sustainability Plans with high integration scores, we

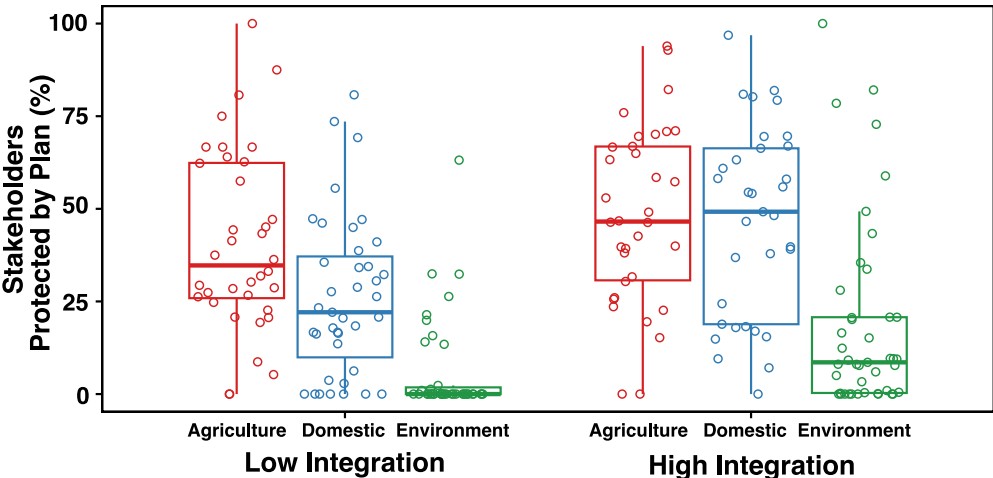

**Fig. 4 | Boxplots of protection values for agriculture (agricultural wells), domestic (domestic wells), and environment (groundwater-dependent eco-systems) groups for plans where stakeholder integration was low (left) and high (right).** The boxplots show the 25th, 50th and 75th percentile of actual pro-tection values (depicted as circle markers) for each stakeholder within low inte-gration and high integration groups. For all groups, protection was greater when integration was high compared to when integration was low. When integration was low, we found significant differences among all group's protection values. When integration was high, protection for domestic and agriculture groups was similar (i.e., non-significant difference), and protection for the environment group was significantly lower than the domestic and agriculture groups.

also found significant differences among mean protection scores ($F = 17.19$, $p \le 0.001$), but pairwise mean protection scores for the domestic and agriculture groups were not distinguishable from each other ($p > 0.05$; Supplementary Table 6.3). Thus, protection values for the domestic and agriculture groups were similar at levels of high integration. Protection scores for the environment group remained significantly lower than for the domestic and agriculture groups at levels of high integration ($p \le 0.001$; Supplementary Table 6.3). Although increased integration is associated with increased protection for the environment, protection lags significantly behind the agri-culture and domestic groups.

To better characterize the relationship between stakeholder integration into a Sustainability Plan and coverage and protection for agricultural, domestic, and environmental stakeholders, we used ordinary least-squares linear models (Fig. 5a–f; Supplementary Table 6.4). We found *aggregate* scores for the domestic and environ-ment groups to be positive and significant predictors of coverage ($\beta_{domestic} = 2.83$, $p_{domestic} \le 0.05$; $\beta_{environment} = 5.50$, $p_{environment} \le 0.01$) and protection ($\beta_{domestic} = 4.34$, $p_{domestic} \le 0.01$; $\beta_{environment} = 2.99$, $p_{environment} \le 0.05$). We found no significant relationship between agriculture's *aggregate* score and its coverage or protection ($p > 0.05$). For the domestic group, a one-point increase in the *aggregate* score was associated with 2.8% greater well coverage and 4.3% greater well protection. Similarly, for the environment group, a one-point increase in the *aggregate* score was associated with 5.5% greater ecosystem coverage and 3.0% greater ecosystem protection.

Additionally, we assessed if each component (i.e., *engage*, *describe*, *analyze*, *act*) independently predicted stakeholder coverage and protection using ordinary least-squares linear models (Fig. 5g–l; Supplementary Tables 6.5–6.10). Coverage and protection for agri-cultural stakeholders, as well as coverage for domestic stakeholders, were not significantly influenced by individual integration component scores. Nevertheless, for domestic stakeholders, a high score for *analyze* and *act*, compared to a low score, was associated with 15% ($\beta = 7.47$, $p \le 0.05$) and 16% greater well protection ($\beta = 8.19$, $p \le 0.01$), respectively. For environmental stakeholders, a high score for *engage*, compared to a low score, was associated with 26% greater coverage ($\beta = 12.97$, $p \le 0.01$) and 13% greater protection ($\beta = 6.30$, $p \le 0.05$). Additionally, a high score for *analyze*, compared to a low score, for environmental stakeholders was associated with 16% greater

protection ($\beta = 7.86$, $p \le 0.05$). In short, no integration components are associated with higher coverage or protection for agricultural stake-holders, and no single integration component consistently explains the variation in coverage and protection for both domestic and environmental stakeholders.

Finally, to further our sensitivity analysis for alternative definitions of *engage* for the agriculture group (Supplementary Section 7), we tested if agricultural stakeholders are better protected in plans with higher *aggregate* scores resulting from the engagement of agricultural governing bodies on Groundwater Sustainability Agency boards. There was no relationship between agriculture's *aggregate* integration scores and its coverage and protection (Supplementary Table 7.1), but high *engage* scores were significantly associated with lower coverage (Supplementary Table 7.2). In short, when we account for agricultural governing bodies, we still do not find any integration components important for protection, but for the first time in our assessment, an integration component (*engage*) is negatively associated with coverage.

## Economically vulnerable groups are less integrated and protected

We considered two subgroups − small farms and disadvantaged com-munities − that fall within the agricultural and domestic groups, respectively. These subgroups are considered economically vulnerable to groundwater depletion because of the costs[18] required to adapt to declining groundwater levels. Small farms are defined as farms with an annual gross cash farm income of less than $350,000[17], and dis-advantaged communities are designated under CA Water Code 79505.5 as communities with "an annual median household income that is less than 80 percent of the statewide annual median household income".

We evaluated and statistically compared integration components and *aggregate* scores between subgroups and their respective stake-holder group using Mann−Whitney Wilcoxon Rank-Sum tests (Fig. 6, Supplementary Table 2.4, Supplementary Figs. 2.2 and 2.3). Small farms scored significantly lower than agriculture for *engage* ($p \le 0.001$), *describe* ($p \le 0.001$), *analyze* ($p \le 0.001$), *act* ($p \le 0.001$), and the overall *aggregate* score ($p \le 0.001$). Disadvantaged commu-nities scored significantly lower than domestic for *describe* ($p \le 0.001$), *analyze* ($p \le 0.001$), *act* ($p \le 0.05$) and the overall *aggregate* score ($p \le 0.01$; Supplementary Fig. 2.4). In conclusion, subgroups that are

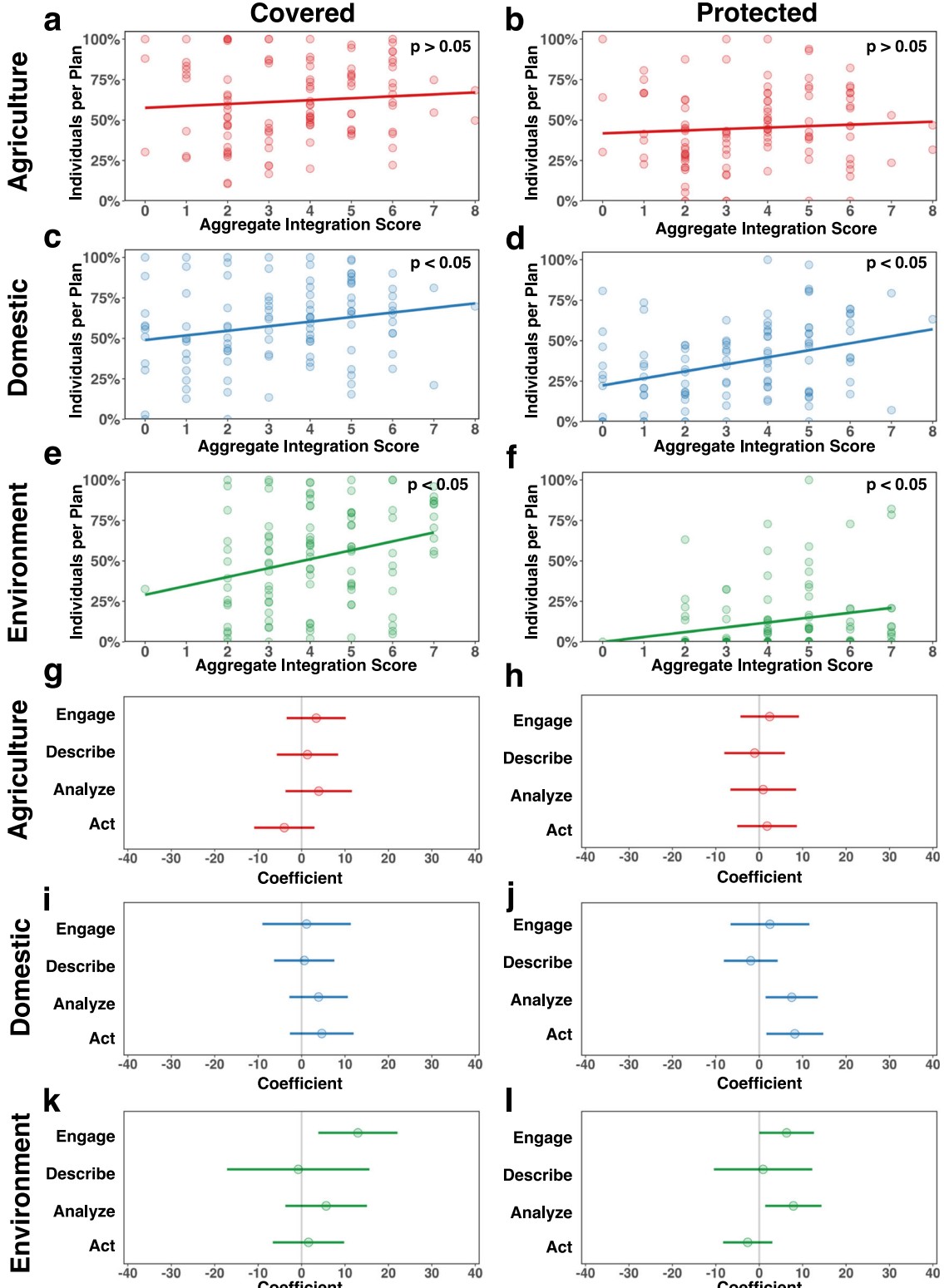

**Fig. 5 | Stakeholder integration and its influence on coverage and protection.** **a–f** Scatter plots showing the relationship between integration *aggregate* scores and percentage covered and protected by stakeholder group, including the ordinary least-squares linear fit and *p*-value. **a, b** *Aggregate* integration score for the agriculture group was not significantly associated with coverage and protection ($p > 0.05$). **c, d** *Aggregate* integration scores for the domestic group were positive and significant predictors of coverage ($\beta = 2.83$, $p \leq 0.05$) and protection ($\beta = 4.34$, $p \leq 0.01$). **e, f** *Aggregate* scores for the environment group were positive and significant predictors of coverage ($\beta = 5.50$, $p \leq 0.01$) and protection ($\beta = 2.99$, $p \leq 0.05$). **g–l** Coefficient dot-and-whisker plots showing which integration

component(s) predicted whether a stakeholder group was covered or protected. Dot and whiskers that entirely fall on either side of a coefficient of zero are considered significant ($p \leq 0.05$). **g, h** Agriculture's coverage and protection were not significantly influenced by the *engage, describe, analyze,* or *act* scores. **i, j** Domestic's coverage was not significantly influenced by the *engage, describe, analyze,* or *act* scores, but protection was positively and significantly associated with *analyze* ($\beta = 7.47$, $p \leq 0.05$) and *act* ($\beta = 8.19$, $p \leq 0.01$) scores. **k, l** Environment's coverage was positively and significantly associated with *engage* scores ($\beta = 12.97$, $p \leq 0.01$), and protection was positively and significantly associated with *engage* ($\beta = 6.30$, $p \leq 0.05$) and *analyze* scores ($\beta = 7.86$, $p \leq 0.05$).

considered more economically vulnerable to groundwater depletion are less integrated, generally, than their respective stakeholder groups.

We analyzed coverage and protection by each Sustainability Plan's groundwater level minimum thresholds established at representative monitoring wells for those within communities designated as disadvantaged ($n_{designated}$ = 29,405 wells) and not designated as disadvantaged ($n_{not\_designated}$ = 58,349 wells). Wells associated with the small farms subgroup were not assessed, because geospatial data identifying the location of small farms were not available. Forty-eight percent ($n_{covered}$ = 14,194) of wells within communities designated as disadvantaged and 49% ($n_{covered}$ = 28,522) of wells within communities not designated as disadvantaged were covered (Supplementary Table 4.1). Thirty-four percent ($n_{protected}$ = 10,106) of wells within communities designated and 38% ($n_{protected}$ = 22,343) of wells within communities not designated as disadvantaged were protected. We found no significant difference in well coverage between the two groups ($p > 0.05$), but found domestic wells within disadvantaged communities to be significantly less protected than domestic wells outside these communities ($p \leq 0.001$; Supplementary Table 4.2). In short, domestic wells within designated disadvantaged communities are less protected than domestic wells outside designated disadvantaged communities.

*Aggregate* integration scores for communities designated as disadvantaged were overwhelmingly low (Supplementary Fig. 2.3). As a result, variation was not sufficient to create tertiles to assess the relationship between stakeholder integration and protection for disadvantaged communities. Using ordinary least-squares linear models, we found that stakeholder integration for disadvantaged communities was not predictive of coverage or protection. This was the case for the *aggregate* score and the individual integration components (Supplementary Tables 6.11–6.13, Supplementary Fig. 6.2).

## Stakeholder decision-making hinges upon local discretion in the absence of state directives

Groundwater depletion is a major threat to food systems, communities, and ecosystems around the globe. California has long been a hotspot of groundwater depletion[1], and the recent passage of California's Sustainable Groundwater Management Act creates an unprecedented opportunity to evaluate the role of stakeholder integration in protecting agricultural, domestic, and environmental stakeholders from declining groundwater levels. Overall, we find 91% of the 108 Sustainability Plans fail to comprehensively *address each of the integration components for all* stakeholders.

In the name of local control and flexibility, California's Sustainable Groundwater Management Act is a state-mandated participatory process[19] that gives discretion to local Groundwater Sustainability Agencies in how they integrate stakeholders. Drawing from the literature on collaborative governance and planning theory, planning processes are described as tiered stages through which a problem is defined, approaches to addressing the problem are identified, and then a decision is made to select the approach(es)[9,20]. As such, we expected that the *engage, describe, analyze,* and *act* components would progress sequentially, representing coordinated integration of local knowledge and social learning through stakeholder participation[8,12]. Instead, our results suggest that stakeholder integration in the majority of plans was not a sequential, coordinated process. Although domestic and environmental stakeholders are well described and considered with respect to potential groundwater loss (i.e., through high *describe* and *analyze* scores), their needs are not addressed in plan actions (i.e., explicit benefits from management actions captured in *act* scores). Alternatively, agricultural stakeholders achieve the same level of support through plan actions (*act*) as the environmental stakeholders, regardless of agriculture's *engage, describe,* and *analyze* scores. Our results reflect varied incentives and motivations to address stakeholder needs, and are in some cases likely the result of more explicit regulatory

requirements (e.g., *describe* for the environment as shown in Supplementary Table 2.1), in other cases due to local dynamics that prioritize one or more groundwater uses over others, or both.

Although some stakeholder groups excelled at some components, we found no compelling evidence that any stakeholder group is more explicitly integrated in *aggregate* than another; this includes when we account for agricultural stakeholders' differential ability to be represented by governing entities, such as reclamation and irrigation districts, on Groundwater Sustainability Agencies boards. Low integration of all stakeholders into plans is likely related to California's Sustainable Groundwater Management Act's regulatory language[21] which prescribes processes for local entities to define and establish sustainable management criteria without mandating protection outcomes specific to stakeholder groups. Alternate models of stakeholder-related provisions in natural resources regulation show that performance-based regulation (i.e., specifying the outcome to be achieved, rather than how to achieve it) is possible: e.g., developing water plans that must not negatively impact the current levels of protection for specified stakeholder groups (Basin Plan 2012 (Austl.) section 10.54). Such an alternative model could be especially useful for the vulnerable subgroups explored here. Although no single stakeholder group is more integrated than another, our results indicate that the two subgroups – small farms and disadvantaged communities – are less integrated than their respective stakeholder groups. Local-scale governance can incorporate resource users' preferences and valuable local knowledge[22], but when directive is low and discretion is high, there is a greater risk that Sustainability Plans may be co-opted by economically powerful local interests at the expense of diverse voices[20,23–27].

## Protection is unequal and vulnerable groups experience greater burdens

Our results raise questions about definitions of sustainability – ensuring that societal, ecological, and economic needs of all user groups are protected – in Sustainability Plans. California's Sustainable Groundwater Management Act does not require that Sustainability Plans mandate protection outcomes specific to stakeholders. Consequently, and perhaps unsurprisingly, our results highlight that >75% of Sustainability Plans do not define minimum thresholds in a way that explicitly protects stakeholders or minimizes impacts to stakeholders. The repercussions of this play out clearly: 60% of agricultural wells, 63% of domestic wells and 91% of groundwater-dependent ecosystems in California's regulated basins are not protected from losing access to groundwater based on each Sustainability Plan's stated minimum thresholds.

At the extreme, unprotected ecosystems could be lost and unprotected wells could run dry – both phenomena are already happening across California. When ecosystems are lost, there is a cascading effect on other public goods, such as protecting critical habitat for threatened and endangered species, and ecosystem services, like water purification, flood mitigation, and climate regulation[28–31]. When wells run dry, people lose access to water[18,32–34], jeopardizing farming operations and depriving people of water for household needs. Deeper wells can reduce the immediate impacts of declining water levels, but constructing new wells and deepening existing wells is expensive and requires pumping water from deeper depths, escalating acquisition costs. Moreover, there is increasing evidence that wells that are running dry are clustered in rural, low-income communities[34] that may have limited access to alternative water sources. When we explored protection for economically vulnerable subgroups, we found that domestic wells inside designated disadvantaged communities are significantly less protected than domestic wells outside these communities. Burdens placed on stakeholders that rely on shallow groundwater (e.g., households, ecosystems) or stakeholders that are economically vulnerable (e.g.,

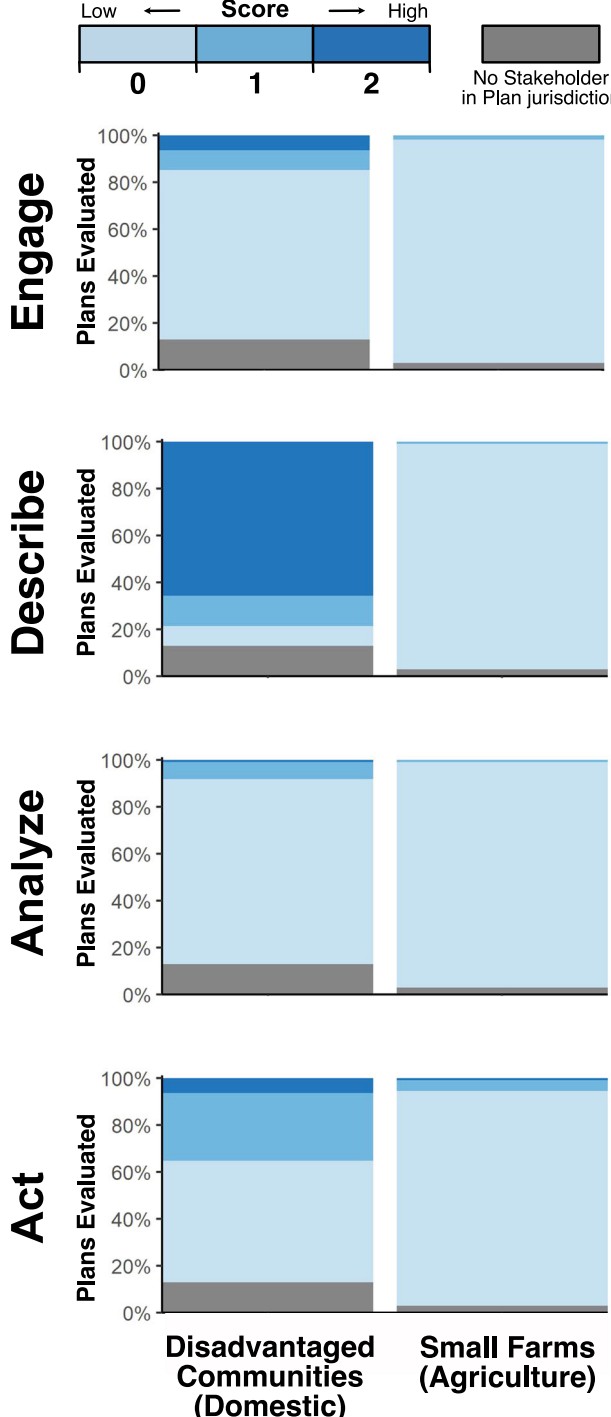

**Fig. 6 | Stakeholder integration scores for sub-groups: disadvantaged communities and small farms.** Integration scores range between zero and two for each integration component: *engage, describe, analyze*, and *act*. For each sub-group, subplots show the percentage of Sustainability Plans evaluated that received a zero, one, or two score within each integration component. Sustainability Plans without a particular subgroup are depicted in gray. In general, integration scores for disadvantaged communities and small farms were overwhelmingly low. When compared with the major stakeholder groups in Fig. 2, in general, disadvantaged communities scored lower than the domestic group, and small farms scored lower than the agriculture group. Two exceptions were disadvantaged communities' *engage* score, which was similar to the domestic group, and disadvantaged communities' *describe* score, which was higher than the domestic group.

small farms, disadvantaged communities) intensify the urgency of ensuring their integration into decision-making processes, so that their water needs are considered and addressed in planning and implementation.

## Diverse stakeholder integration predicts better stakeholder outcomes

Despite low levels of stakeholder integration and stakeholder protection in the Sustainability Plans overall, our results demonstrate two key findings: (1) agricultural stakeholder integration is not linked to protection, and (2) when domestic and environment groups are integrated, outcomes are better for those groups.

Agricultural users have the highest level of protection among stakeholder groups at both low and high levels of integration. As such, our results suggest that invited and visible stakeholder representation for agriculture, the focus of our analysis, is less relevant for agriculture's outcomes than for the other groups. These results are unsurprising. Case study evidence suggests that farmers feel confident in their Groundwater Sustainability Agency representation[35] and interview data with participants in local Sustainability Plan processes have found evidence of agricultural interests dominating Sustainability Plan processes[23,26,36]. Agricultural users often are represented by agricultural entities within Groundwater Sustainability Agencies. These entities include reclamation and irrigation districts, or more broadly, entities with formal governance authority. Here, formal governance authority means that agricultural stakeholders can participate in the process on their own accord, i.e., without needing to be invited by others with power.

The literature on power in decision-making processes[37] highlights pathways for hidden power dynamics, or ways in which some stakeholders may wield power in forms that may not be easily observable. For example, agricultural governing bodies can execute power in setting the Sustainability Plan agenda, deciding priorities and processes for Sustainability Plan development, and deciding whether to invite other stakeholders on advisory committees or board voting seats. In our sensitivity analysis we found evidence for what may be agricultural governing bodies executing this less visible power: Sustainability Plans with agricultural governing bodies within Groundwater Sustainability Agencies are associated with less coverage for agriculture. Less coverage for agricultural wells means that there are fewer representative monitoring sites where agriculture is pumping groundwater, which could lead to undetected undesirable results.

In contrast to agriculture, opportunities for domestic and environmental groups to influence decision-making processes are limited despite the regulatory requirement to consider these users in Sustainability Plans. Often, stakeholder groups that lack formal governance authority to participate, such as domestic and environmental groups, also lack financial resources, time, and technical capacity to engage meaningfully in the planning process[32,38]. Early stakeholder engagement in the policy processes is important, but it is also key to consider how and when to engage stakeholders throughout the entirety of the process[39]. Without the ability to form a governing body to represent their interests, the most straightforward route for domestic and environmental stakeholders to influence decision-making processes is to be appointed stakeholder seats on governing boards.

Our results suggest that invited and visible stakeholder integration for domestic and environmental groups is an important avenue towards improved outcomes for these stakeholder groups. This is not to say that it is sufficient to produce better outcomes for these user groups. In some cases, stakeholder engagement can be a form of token participation that gives the facade of diverse inclusion, but lacks an ability to sway hegemonic power relations or majority voting rules[40].

For domestic stakeholders, higher levels of integration were associated with similar levels of protection to agricultural stakeholders. Although protection for environmental stakeholders increased with higher levels of integration, environment's protection still lagged domestic and agricultural stakeholders significantly. Surprisingly, we find no evidence that disadvantaged communities benefit from stakeholder integration, but this finding is likely because no plan had high integration scores for disadvantaged communities. Without examples of high integration for disadvantaged communities, we cannot statistically interpret this result to suggest that integration is not important for disadvantaged communities; based on the results from the domestic and environment groups, it is likely that integration is important for disadvantaged communities. More research is needed to understand the drivers and processes of equity in groundwater governance processes[41,42].

## Sustainability plans epitomize global trends in water planning and stakeholder integration requirements

Overall, for the majority of Sustainability Plans, stakeholder integration is not a sequential, coordinated process, but for the few Sustainability Plans that integrate comprehensively, domestic and environmental stakeholders are more protected. Previous literature emphasizes that successful stakeholder engagement is built upon a foundation of fairness, trust, respect, and co-learning[19,39]. Our approach reflects on these aspects of integration into Sustainability Plans as a proxy for stakeholder integration into the local policy processes that resulted in the 108 Sustainability Plans by evaluating four components: *engage*, *describe*, *analyze*, and *act*. The stakeholder integration literature has focused a great deal on the specifics of the policy process that lead to better integration[11]; we build on this here to demonstrate that when integration happens, it leads to better outcomes for stakeholders that lack formal governance authority. In short, when the four integration components are taken together, the needs and knowledge of stakeholders can become better represented into planning and policy outcomes, which can lead to better outcomes among diverse user groups.

Globally, governments are moving away from unmanaged natural resources and towards long-term planning arrangements that include local management with and engagement of diverse user groups[6,7]. The trend towards stakeholder engagement offers promise for better reflecting the needs of diverse user groups, including disadvantaged communities, small farms, and ecosystems. Nevertheless, the global experience, like California's experience, reveals many barriers to translating these policy trends into substantive outcomes, including token forms of participation that fail to influence decisions, corruption, lack of resources, and factors that centralize authority in those with economic power (e.g., Refs. [40,43–45]). Sustainability policies are on the rise to address some of society's toughest challenges – climate change, biodiversity loss, and natural resource depletion. Integrating diverse stakeholder voices and needs throughout sustainability planning and management can provide better outcomes among stakeholders, especially those who are most vulnerable to the impacts of natural resource depletion.

## Methods

### Evaluation of stakeholder integration into groundwater sustainability plans

Individual stakeholder groups and subgroups (i.e., agriculture, domestic, environment, disadvantaged communities, small farms) were evaluated to quantify stakeholder integration in each Sustainability Plan. These groups are not representative of all interested parties or rights holders, including Native American Tribes (e.g., federally and state recognized tribes, and unrecognized federally and state indigenous communities), or stakeholders under the Sustainable

Groundwater Management Act, including municipal water districts or industries. Specifically, we focused on individual users that self-supply (or self-access) groundwater rather than corporations, agencies, or tribes that have governing bodies to manage their water resources. For each stakeholder group, we evaluated Sustainability Plan text and spatial data to determine stakeholder group presence or absence for inclusion in our analysis (Supplementary Section 2; Supplementary Tables 2.2–2.3).

We evaluated stakeholder integration for each group based on four components: *engage*, *describe*, *analyze*, and *act* (Supplementary Section 2). Broadly, these integration components captured the Sustainability Plan requirements under California's Sustainable Groundwater Management Act: from inviting stakeholder participation, describing stakeholder needs, considering potential impacts of groundwater conditions on stakeholders when establishing sustainable management criteria, and supporting stakeholder needs through planned actions (Supplementary Table 1.1). Each construct was evaluated through a series of questions on an evaluation rubric that was applied to each Sustainability Plan. The final evaluation rubric consisted of five questions: two binary *engage* questions for presence of a designated stakeholder seat on an advisory committee and voting board that were combined for the score, and a single question each for *describe*, *analyze*, and *act*. These questions were evaluated for all three stakeholder groups (i.e., domestic, agriculture, environment) and both subgroups (i.e., disadvantaged communities, small farms; Supplementary Table 2.1). Each component featured a three-level ordinal numerical response spectrum (i.e., no: 0; somewhat: 1; yes: 2).

We reviewed two batches of Sustainability Plans: 45 critical basin Sustainability Plans submitted in 2020, and 63 high and medium priority Sustainability Plans submitted in 2022 (Supplementary Table 2.2). This research was built upon previous NGO-led evaluations of the 2020 Sustainability Plans[46,47]. Each Plan was evaluated during the corresponding public comment periods; the data collected from the Sustainability Plans was also used to inform public comment letters. Our review of Sustainability Plans did not include an evaluation of revised 2020 Sustainability Plans that were deemed "incomplete" by the California Department of Water Resources, since we wanted to compare stakeholder integration and protection across both 2020 and 2022 Sustainability Plans at the same planning stage, and 2022 Sustainability Plans had not yet been evaluated by Department of Water Resources at the time of submission. In total, there were three rounds of data collection (coding): (1) draft 2022 Sustainability Plans the Groundwater Sustainability Agencies released during May-December 2021 for public comment; (2) final 2022 Sustainability Plans submitted to the California Department of Water Resources in January 2022; and (3) re-coding of the final 2020 and final 2022 Sustainability Plans with our final, refined set of integration questions. Each round of coding included five to seven individual coders working concurrently on evaluating the Sustainability Plans. In each round, an individual coded all Sustainability Plans for a subset of questions to ensure consistent application of the rubric questions. The one exception to this was for the *engage* questions in the third round of coding, which was coded by two coders for an additional accuracy check. In this case, the two coders each coded the *engage* questions for all Sustainability Plans, compared responses at regular intervals to ensure Cohen's kappa for intercoder reliability was above 0.7 representing substantial agreement[48], and reconciled responses. Additionally, each round of coding included a quality control review by one of the first authors, in which all coding for each Sustainability Plan was reviewed to ensure accurate implementation of the rubric.

Additionally, we conducted a sensitivity analysis for the agriculture group's *engage* component. Agricultural stakeholders have the differential capacity of being represented by governing entities

within Groundwater Sustainability Agency boards; domestic and environmental stakeholders do not have this capacity. Although our paper focuses on individual stakeholders, we recognized that agricultural stakeholders have additional power through their governing entities. For our sensitivity analysis, we defined two variations of the *engage* integration component for the agriculture group that capture the presence of a Groundwater Sustainability Agency specific to agriculture on the board (Agriculture 2) and the presence of both an agricultural entity and designated agricultural seat on the governing board (Agriculture 3; Supplementary Section 3).

## Spatial analyses to quantify coverage and protection of stakeholders

Within each Sustainability Plan's boundary (https://sgma.water.ca.gov/portal/#gsp, accessed 25 May 2022), we identified all representative monitoring wells ($n$ = 5204 monitoring wells) used to establish minimum thresholds for the groundwater level sustainability indicator (Supplementary Section 4). Representative monitoring well locations and their minimum thresholds were downloaded from the California Department of Water Resources's Sustainable Groundwater Management Act portal (https://data.cnra.ca.gov/dataset/gspmd, accessed 7 June 2022), which were submitted by each Groundwater Sustainability Agency to California's Department of Water Resources.

We analyzed the location and minimum thresholds of each monitoring well relative to each stakeholder within a stakeholder group within a Plan boundary ($n_{agriculture}$ = 37,869 wells; $n_{domestic}$ = 87,754 wells; $n_{environment}$ = 1540 km² groundwater-dependent ecosystems; $n_{designated}$ = 29,405 domestic wells in designated disadvantaged communities). Agricultural and domestic stakeholders were identified using groundwater well data downloaded from the California Department of Water Resources' Online System of Well Completion Reports (OSWCR; https://water.ca.gov/Programs/Groundwater-Management/Wells/Well-Completion-Reports, accessed 23 June 2022). We selected wells with a recorded construction date from 1975 to 2022 to account for the retirement of older wells. We selected wells with the record type "WellCompletion/New/Production or Monitoring/NA", which best represents the construction of new wells. We selected only records with a reasonable depth (i.e., depth >0) and reasonable location (i.e., within California). For domestic wells, we selected wells with "Water Supply Domestic" as the recorded use; for agricultural wells, we selected wells with "Water Supply Irrigation - Agriculture", "Water Supply Irrigation - Agricultural", "Water Supply Irrigation Agricultural", and "Water Supply Irrigation Stock or Animal Watering" as the recorded use. The database includes duplicate records; we selected records with distinct entries for well record number, latitude, longitude, well depth, construction date, and use.

Disadvantaged communities (i.e., defined by California Water Code 79505.5 as communities with an annual median household income <80% of California's annual median household income) were identified using census block data (2018) downloaded from California's Disadvantaged Communities mapper tool (https://gis.water.ca.gov/app/dacs/, accessed 16 June 2022). The disadvantaged communities' subgroup includes domestic wells within census blocks designated as disadvantaged communities. Small farms were not spatially analyzed because spatial data delineating the locations of small farms is not available at a disaggregated scale.

Environmental stakeholders (i.e., groundwater-dependent ecosystems that contain species and ecological communities reliant on groundwater occurring near or on the ground surface) were identified using the California Department of Water Resources' Natural Communities Commonly Associated with Groundwater dataset[49], which maps vegetation, wetlands, springs, and seeps likely to be associated with groundwater. All rooting depths were accessed from The Nature Conservancy's Plant Rooting Depth database (https://www.groundwaterresourcehub.org/where-we-work/california/plant-rooting-depth-database/, accessed 14 June 2022). The rooting depths were corrected for land surface elevation using the United States Geological Survey 3DEP 10 m resolution dataset (https://apps.nationalmap.gov/downloader/, accessed 13 June 2022), so rooting depth elevations could be compared with the groundwater elevation minimum thresholds.

We used a total of five geospatial buffers – i.e., ~0.8 km (0.5 mi), ~1.6 km (1.0 mi), ~2.4 km (1.5 mi), ~3.2 km (2.0 mi), ~4.0 km (2.5 mi) – to conduct a sensitivity analysis to assess how the buffer size influenced the coverage and protection results. A buffer represents a circular zone determined by a horizontal distance (~2.4 km) from each representative monitoring well. Within each Sustainability Plan boundary, we calculated the total amount of area covered when using each of the five buffers. We used histograms representing the percentage of every Sustainability Plan area covered by the total buffered area surrounding representative monitoring wells to select the most appropriate buffer size for our analyses (Supplementary Fig. 4.1). The ~0.8 and ~1.6 km buffers skewed left (i.e., most plans had less than 50% of their area covered) and the ~1.6 and ~2.4 km buffers skewed right (i.e., most plans had more than 50% of their area covered). The ~2.4 km buffer was the closest to a normal distribution and was selected for the analysis, because this buffer was least prone to skewed results and considered a reasonable distance to reflect local groundwater conditions.

To assess whether stakeholders were covered spatially by any representative monitoring well, we applied a ~2.4 km buffer around each representative monitoring well (Fig. 1c, d). Any portion of a representative monitoring well's buffer that was outside the representative monitoring well's Sustainability Plan boundary was removed to ensure that only stakeholders within a given Sustainability Plan boundary were assessed. If a stakeholder (i.e., agricultural well, domestic well, or wells within areas designated as a disadvantaged community) or any portion of a stakeholder (i.e., groundwater-dependent ecosystem) was located within the monitoring well's buffer, it was designated as being covered (Fig. 1c, d). If a stakeholder was covered, we then assessed if the associated representative monitoring well's minimum threshold was protective of the stakeholder. If multiple monitoring wells covered a stakeholder, we assessed whether any of the monitoring wells protected the stakeholder. Stakeholders were only counted once, even if multiple monitoring wells covered and protected them. For covered agricultural and domestic stakeholders and stakeholders within the disadvantaged communities subgroup, protection occurred when the respective stakeholder's total well depth was deeper than the minimum threshold established at the nearby monitoring well. For ecosystems, protection occurred when the maximum rooting depth for vegetation (Supplementary Table 4.7) was deeper than the minimum threshold established at the nearby monitoring well.

## Statistical analyses

All data analysis and statistical tests were performed using the programming language R[50]. We created an overall *aggregate* integration score for each stakeholder group per Sustainability Plan by summing the scores for the *engage*, *describe*, *analyze*, and *act* components. To compare differences between agricultural, domestic, and environmental stakeholder groups on the four integration components and the *aggregate* score, we ran Kruskal–Wallis rank sum tests in the base R stats package (*kruskal.test*), followed by Dunn post-hoc test for pairwise comparison while controlling for multiple comparisons using the Fisheries Stock Analysis (FSA) package in R[51]. To test for differences between stakeholder groups and subgroups, namely agriculture-small farms and domestic-disadvantaged communities, we ran Wilcoxon Rank Sum tests using the base R stats package (*wilcox.test*).

To test hypothesized relationships between integration scores and coverage and protection data for wells and ecosystems, we used

one-sided *t*-tests in the base R stats package (*t.test*) to compare the mean protection score for the lowest versus highest tertile of *aggregate* integration scores by stakeholder group. These groups approximated the lowest and highest tertiles, while keeping like integer scores together in groups. To compare between stakeholder groups, we ran Analysis of Variance tests (*aov*) with Tukey's pairwise comparison (*TukeyHSD*) in the base R stats package to examine differences in mean protection scores in the lowest and highest tertiles among stakeholder groups.

To better characterize the relationship between integration data and well and ecosystem coverage and protection data, we ran six Ordinary Least Squares linear regressions in the base R stats package (*lm*) between each of the three stakeholder groups' integration score and the corresponding coverage and protection percentages. We then tested the relative predictive power of each independent integration component, by running six additional Ordinary Least Squares linear regressions using the base R stats package with the four integrate components (i.e., *engage*, *describe*, *analyze*, and *act*) as independent variables and the same corresponding covered and protected percentages per Sustainability Plan as dependent variables.

In addition, to test if our results were sensitive to the ability of agricultural stakeholders to be represented by agricultural governing entities within Groundwater Sustainability Agencies, we ran the same set of statistical analyses with the two variations in the coding of *engage* for the agriculture group (Supplementary Section 7).

### Limitations

California's Sustainable Groundwater Management Act requires that Sustainability Plans address six undesirable results (Supplementary Section 1). Our assessment looked at only one of the six undesirable results: groundwater level declines. The performance of the Sustainability Plans on other criteria might be better if stakeholders were pushing for action on other criteria.

Our analysis of well protection used total well depth, because pump depth and screening interval depths were not available comprehensively across the Sustainability Plan jurisdictions. Because we used the well depth, our analysis provides a more conservative estimate of well protection than if we used pump depth or screening interval depths. Additionally, well construction data for groundwater wells only provided location data to the centroid-of-sections for many wells. To assess protection, we calculated the elevation of the bottom of the well using land surface elevation data to compare it with the minimum threshold elevation at the associated representative monitoring well. For wells with centroid of section locations, we calculated the mean land-surface elevation of the section, and for wells with GPS locations, we identified the land-surface elevation at the GPS location.

Sustainability Plans reflect only the stated and visible priorities of the Groundwater Sustainability Agencies, and therefore, are not representative of all of the interactions and power dynamics within the policy process. As such, omitted variable bias was a potential challenge for our models. Local-level, observational variables that were not formalized in the Sustainability Plan were outside the scope of our study. Given the sample size and lack of hypothesized alternative predictors, we focused our study on stakeholder integration and its influence on stakeholder outcomes (i.e., coverage and protection). Nevertheless, we ran additional post-hoc exploratory analyses, using t-tests and ordinary least squares models, to examine whether stakeholder integration or outcomes varied with certain variables, such as the submission year of the Sustainability Plan (2020 versus 2022) and the number of individuals (i.e., total number of wells or ecosystem area) per Sustainability Plan (Supplementary Section 8). From these additional analyses, we found (1) integration scores and coverage and

protection were higher in the Sustainability Plans submitted in 2022, and (2) for each stakeholder group, the number of wells or acres is significantly and negatively associated with coverage, suggesting that as the number of wells and size of a Sustainability Plan area grows, the representative monitoring networks do not scale to cover the increased number of stakeholders within a Sustainability Plan.

## Data availability

All data generated in this study have been deposited in Zenodo (https://doi.org/10.5281/zenodo.7908803)[52]. The raw data are publicly available and are accessible from the persistent web-links provided in the Methods section and provided in the Zenodo database.

## Code availability

All code generated in this study has been deposited in Zenodo (https://doi.org/10.5281/zenodo.7908803)[52]. All code was written using R statistical software (version 4.3.0). Composite figures were assembled in Affinity Designer (https://affinity.serif.com/en-us/designer/).

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

## Acknowledgements

This work was funded by the Water Foundation (all authors). Some material is based upon work supported by the U.S. Geological Survey (USGS) through the California Institute for Water Resources (CIWR) under Grant/Cooperative Agreement No. G21AP10611 (DP). The views and conclusions contained in this document are those of the authors and should not be interpreted as representing the opinions or policies of USGS/CIWR. Mention of trade names or commercial products does not constitute their endorsement by USGS/CIWR.

## Author contributions

These authors contributed equally: DP, MMR, and CHW. DP, MMR, and CHW developed the research program, DP, MMR, CHW, RA, MB, LE-K, MGF, KH, WDJ, SKA, and KAG collected data. DP, MMR, CHW, and CM analyzed data. DP, MMR, CHW, and RN wrote the paper. SA, NA, JPO-P, EJR, and RN provided intellectual support.

## Competing interests

The authors declare no competing interests.
