## [Peer Review File · Nature Communications]

Stakeholder integration predicts better outcomes from groundwater sustainability policyREVIEWER COMMENTS

Reviewer #1 (Remarks to the Author):

This is an excellent submission and I recommend that it be published in its current form. The question being addressed is important -- whether successful efforts to integrate diverse stakeholders into resource management processes are associated (positively) with protection of those stakeholders' interests in the management plans that result from those processes. There are three particularly noteworthy results. First, most of the groundwater sustainability plans the authors evaluated did not successfully integrate stakeholders into the plan development processes. Second, among the minority of sustainability plans that had successfully integrated stakeholders, those stakeholders' interests and vulnerabilities were more likely to be taken into account and protected. Third, the most vulnerable stakeholder groups are also least likely to have been integrated in the plan development processes. The evidence presented supports these results, and I did not perceive any errors or flaws in the methods used. The methods are straightforward -- plans were evaluated for indications of stakeholder integration, a spatial framework was employed for identifying whether stakeholders' wells were covered by the monitoring practices and management thresholds in the sustainability plans, and a regression analysis was employed to assess the presence and direction of any relationship between the data on stakeholder integration and the data on protection of stakeholders' wells. The methods section provides enough information to provide assurance of the soundness of the work, and supplementary information is sufficient to allow for replicability assessment. I commend the authors especially for the clarity of the presentation. The manuscript is eminently readable and interesting, and the value of its contribution to the literature is evident.

Reviewer #2 (Remarks to the Author):

The manuscript "Stakeholder integration predicts more equitable groundwater sustainability policy" utilizes extensive primary and secondary data sources to consider how stakeholder integration effects protection for three sets of stakeholders from declining groundwater levels under California's Sustainable Groundwater Management Act. The underlying research endeavor is impressive in its breadth and level of detail and the case is undeniably important and informative. As such, I sincerely look forward to seeing the results published hopefully in multiple products. Nonetheless, I have some concerns about the analytical approach and framing and the respective implications for the paper's findings and contributions. Below I detail these concerns followed by some more specific comments and suggestions by line item.

First and foremost, I am concerned by the comparison of two stakeholder groups that lack "institutional representation" (environment and domestic wells) to a group that has significant "institutional representation" (agriculture). I believe this has serious consequences for interpreting the results. For example, the authors assert they find no evidence that any stakeholder group is more integrated than others, but this finding is in direct contradiction with significant peer-reviewed research on this case. Subsequently, the authors posit that stakeholder integration does not have a significant effect on protection for agriculture, but I don't see how you can come to this conclusion given that agriculture is highly integrated, just in a different manner. In another case, if agriculture did not have institutional representation, stakeholder integration per the authors definition may indeed have an impact. I do not think these two elements of representation can be discussed separately, even more so if regulatory decision-making authority (seat on board) is included as part of the engage scoring criteria as it appears to be. I don't see how it can be argued that a seat on a GSA board for an irrigation district whose whole mission is to serve farmers and whose board is entirely made up of farmers is substantively different than an appointed seat for an environmental representative. Both feel equally pertinent to who loses out under the plan.

I am also a bit unconvinced by the deployment of equity as a focal concept for the paper. Lines 80-82 provide the authors definition of equity as how similarly each user group is protected. The manuscript asserts that stakeholder integration enhances equity for all stakeholders, but the

results clearly show that ecosystem protection lags far behind agriculture and domestic stakeholders. To me the results show improved outcomes for the stakeholder groups, not equity. If the authors wish to keep equity in the paper, I think there is need for more nuance and complexity in the discussion, especially since less disparities in protection levels between these groups doesn't account for variable impacts of those levels as the authors discuss in the discussion. Relatedly, not including DACs in the analysis of the effects of integration feels like a major missed opportunity to more meaningfully get at equity in the paper as I discuss further in the specific comments below.

Third, I have several questions and concerns about the stakeholder integration score. That environment has the highest aggregate score seems odd based on existing literature about SGMA. As you note this is driven by the high describe component score. This, in turn, I suspect is driven by the requirement to map groundwater dependent ecosystems noted in supplementary table 2.1. If the other stakeholders don't have similar, specific requirement, I suspect this finding is picking up on this requirement (which based on the results seems relatively ineffective) rather than reflecting stakeholder integration and really confounds interpretation. The extent to which similar statutory requirements could be affecting other components should also be considered. Related to the "act" component, I have a lot of questions about what was counted for the scoring here and need more information to interpret this element effectively. However, as described, this component feels more akin to an outcome or output rather than stakeholder integration. This needs to be defended as it is odd to use an outcome to predict another outcome. Would we not expect that the two to be correlated?

Lastly, I worry that unobserved variables are driving the OLS results. The fact that no single component explains the aggregate score effect highlights this potential for me. I am sympathetic to the data challenges here but where possible it would be great to try out a few controls, at least maybe the total number of individuals per plan for the category and basin prioritization status. At the very least, the manuscript should include a discussion of this limitation and others.

Specific comments and suggestions

- Abstract line 38: This sentence feels overly specific considering existing literature. I agree that the literature theorizes that integrating stakeholders can provide protections from adverse management impacts but I don't think there is specific literature to this point about groundwater depletions nor do I agree that groundwater depletion is an adverse impact when it comes to all stakeholders. I suggest re-writing in a more general way.
- Abstract missing two of three key findings as defined in the manuscript (lines 101-112)
- I really appreciate the clear and effective summation of findings at the end of the introduction.
- Lines 114-124: Really can't tell from this paragraph what those four components mean. For transparency these components and their scoring should be described in the manuscript not just the supplemental.
- Lines 135- 138: Please clarify what this means.
- Line 199: "as horizontal distances..." Really important to be super clear here and elsewhere this is a methodological/sensitivity analysis point. The material impact on "uncovered" wells is not changed. Again on line 280 as well.
- Lines 204-206: I think this is backwards. As I read SGMA, the intent was to define significant and unreasonable and then define MTs to avoid those conditions.
- Lines 207-209: Can you provide an example of what that looks like. Is it meaningful if not reflected in MTs? As written, it isn't clear to me what this analysis of sustainability goals relates to your research question. I recommend putting this info in conversation with the other findings or delete.
- Line 215-218: Lines 148-157 you talk about measuring integration for these subgroups and you have spatial data on the DACs so I don't understand why DACs are not included. What does including these subgroups in the paper accomplish if they are not part of this analysis? For me, this really limits the equity discussion.
- Lines 229-240: I don't feel like this paragraph does as good of job as it could reflecting the nuances of the results from Fig 4. Would like to see a bit more clarity parsing out the individual score component results.
- In Supplementary table 2.1 "describe" "response levels" What do X/Y and Z mean?
- Did you do any analysis of potential differences based on basin prioritization? Feasibly with less time to write plans there may be a different effect of stakeholder integration.

- I think it would be very helpful to add examples to the supplementary table 2.1 of the response levels for greater transparency/clarity.
- Line 249: clarify here or on lines 125-127 previously if comprehensively means getting at least a one on each of the four or scoring a 2 on each
- Lines 269-271 – Why does this specifically apply to marginalized users and not the three stakeholder categories broadly? I would argue this is why you see such low integration period not just for those two subgroups
- Lines 278-290: I'm not clear how you get from there not being different levels of coverage and protection at the beginning of this paragraph to not protecting domestic and environment groups at the end. Implicitly I wonder if you are getting at the fact that the material impact to these users is likely more severe than to ag (a well-resourced farmer could just drill another well maybe) but this is not clear until the next paragraph. Needs refinement
- Lines 325-236: But regulatory decision-making power is included in your stakeholder integration rubric and scoring criteria under "engage" no? And it is feasible for domestic wells to have regulatory decision-making authority if they were given an assigned seat on a board? I think here and elsewhere you are blurring this concept with institutional representation.
- Line 334-335: What does "integrate" mean here? That the components are correlated?
- For Figure 1 I am confused by panel b. What does "Yes" and "No" mean? Needs clarification.
- Line 459: Five questions but four components? Please explain/clarify
- Line 476: What is step three refined coding? Not clear.
- Are all wells in OSWCR included in the analysis or did you use an estimated retirement age to remove older wells? Needs to be clarified and the implications for the results should be discussed.
- Line 516: It is immediately clear to me why a normal distribution is the deciding factor here? Please clarify this rationale. Does the Department of Water Resources have any guidance on how much spatial area a monitoring well can reasonably represent?
- Lines 128-130: I think it might be better to cite a specific paper you are following here rather than the broader literature. My mind immediately goes to Arnstein's ladder of participation here, but I think the order of these steps would be different if you were following that model. To me engage is a much "more" of an ask than describe or analyze.

Reviewer #3 (Remarks to the Author):

The analysis and findings in this paper offer the field of groundwater management, and natural resource management more broadly, useful quantitative evidence for the benefits of stakeholder integration. As the authors have stated in the manuscript, I am not aware of any other peer-reviewed research that has robustly quantified the impact of stakeholder integration on concrete resource management decision-making. Therefore, these findings are novel as well as useful for practitioners, policymakers, and other researchers.

The approach used to quantify stakeholder integration within the Groundwater Sustainability Plans is robust and well documented. The statistical analysis of the dataset is also well done with no major or minor flaws identified.

The discussion is helpful in addressing the differences identified between agriculture, domestic, and environment. The conclusions do a good job of articulating some of the broader connections that the research has beyond groundwater management in California.

The only suggestions added, in the attached manuscript, pertain to some wording change suggestions and requests for clarity in the choice of terms in a few locations. It may also be worth reviewing literature from the Colorado basin roundtable process for quantitative assessments of stakeholder integration on water management outcomes (if it wasn't reviewed already). Finally, it could be useful to note any additional research questions that the authors believe need to be answered or addressed next.

REVIEWER COMMENTS

This document is a thorough point-by-point response to each comment made by the reviewers. **Each comment made by a review is in blue text. Our responses are in black text.** In places where the reviewer asked for a change that led to substantial edits to the text, we provided the edited/updated text along with line numbers that point to where these edits can be found within the revised manuscript document. In addition to the revised manuscript file, we also provide a file that shows all the changes made between the original manuscript submission and this revised manuscript submission.

TABLE OF CONTENTS

Responses to Reviewer #1 — Page 2
Responses to Reviewer #2 — Pages 3-20
Responses to Reviewer #4 — Pages 21-25

REVIEWER #1 (Remarks to the Author):

This is an excellent submission and I recommend that it be published in its current form. The question being addressed is important -- whether successful efforts to integrate diverse stakeholders into resource management processes are associated (positively) with protection of those stakeholders' interests in the management plans that result from those processes.

We thank Reviewer #1 for their encouraging remarks.

There are three particularly noteworthy results. First, most of the groundwater sustainability plans the authors evaluated did not successfully integrate stakeholders into the plan development processes. Second, among the minority of sustainability plans that had successfully integrated stakeholders, those stakeholders' interests and vulnerabilities were more likely to be taken into account and protected. Third, the most vulnerable stakeholder groups are also least likely to have been integrated in the plan development processes.

We thank Reviewer #1. We also believe that the main results are noteworthy.

The evidence presented supports these results, and I did not perceive any errors or flaws in the methods used. The methods are straightforward -- plans were evaluated for indications of stakeholder integration, a spatial framework was employed for identifying whether stakeholders' wells were covered by the monitoring practices and management thresholds in the sustainability plans, and a regression analysis was employed to assess the presence and direction of any relationship between the data on stakeholder integration and the data on protection of stakeholders' wells.

We thank Reviewer #1. We hope that Reviewer #1 will also find the methods used for the new analyses (in response to Reviewer #2) straightforward.

The methods section provides enough information to provide assurance of the soundness of the work, and supplementary information is sufficient to allow for replicability assessment. I commend the authors especially for the clarity of the presentation. The manuscript is eminently readable and interesting, and the value of its contribution to the literature is evident.

We are glad that Reviewer #1 enjoyed the manuscript and found it to be worthy of publication.

REVIEWER #2 (Remarks to the Author):

Comment 1. The manuscript “Stakeholder integration predicts more equitable groundwater sustainability policy” utilizes extensive primary and secondary data sources to consider how stakeholder integration effects protection for three sets of stakeholders from declining groundwater levels under California’s Sustainable Groundwater Management Act. The underlying research endeavor is impressive in its breadth and level of detail and the case is undeniably important and informative. As such, I sincerely look forward to seeing the results published hopefully in multiple products. Nonetheless, I have some concerns about the analytical approach and framing and the respective implications for the paper’s findings and contributions. Below I detail these concerns followed by some more specific comments and suggestions by line item.

We thank Reviewer #2 for their in-depth review. We are grateful for the time they spent providing specific comments and for the opportunity to address these comments and suggestions prior to publication. We feel that the opportunity to address these comments strengthened the manuscript in both content and clarity. Below, we address each comment in detail.

Comment 2. First and foremost, I am concerned by the comparison of two stakeholder groups that lack “institutional representation” (environment and domestic wells) to a group that has significant “institutional representation” (agriculture). I believe this has serious consequences for interpreting the results. For example, the authors assert they find no evidence that any stakeholder group is more integrated than others, but this finding is in direct contradiction with significant peer-reviewed research on this case. Subsequently, the authors posit that stakeholder integration does not have a significant effect on protection for agriculture, but I don’t see how you can come to this conclusion given that agriculture is highly integrated, just in a different manner. In another case, if agriculture did not have institutional representation, stakeholder integration per the authors definition may indeed have an impact. I do not think these two elements of representation can be discussed separately, even more so if regulatory decision-making authority (seat on board) is included as part of the engage scoring criteria as it appears to be. I don’t see how it can be argued that a seat on a GSA board for an irrigation district whose whole mission is to serve farmers and whose board is entirely made up of farmers is substantively different than an appointed seat for an environmental representative. Both feel equally pertinent to who loses out under the plan.

We appreciate that Reviewer #2 highlighted this important and nuanced dimension of our analysis. To address the comments, we took three steps, described herein, to clarify our language and the definitions of multiple key terms, and conduct new analyses that account for agricultural stakeholders’ ability to have additional engagement opportunities that are not available for the domestic and environment groups.

First, we revised our definition of stakeholder integration to make explicit that we are investigating integration into the written document of the Sustainability Plan. In addition, we clarify that our coding of stakeholder integration is a proxy for stakeholder integration into the local policy processes that resulted in the 108 Sustainability Plans. We also acknowledge that the plans reflect only the stated and visible priorities of the Groundwater Sustainability Agencies, and it is not representative of all of the relative power dynamics within the policy process. Analyzing stakeholder integration in this way gives us the ability to consistently measure and compare stakeholder integration across all 108 Sustainability Plans.

We believe that the aforementioned revisions improve the clarity of our objectives:

- See lines 81-87: We define integration as how well each Sustainability Plan incorporates stakeholders, their knowledge, and needs across four components: *engage*, *describe*, *analyze*, and *act* (Fig. 1b). We use the four components as a proxy for stakeholder integration into the local policy processes that resulted in each of the Sustainability Plans; the plans reflect only the stated and visible priorities of the Groundwater Sustainability Agencies, and not invisible power dynamics that determine decision-making powers within these local agencies.

Second, based on the reviewer's comment, we realized that there was ambiguity in our use of "institutional representation." To clarify our approach, we removed this term. Additionally, we (1) explicitly refer to whether a stakeholder group has the ability to become part of a governing entity, i.e., Groundwater Sustainability Agency, or needs to be "invited" by governing entities to participate, (2) discuss formal governance authority, and (3) differentiate between this formal governance authority that is available to agriculture, and the need for the other stakeholder groups to be invited to participate in the process. Addressing this comment led to significant revisions in our introduction, results, and discussion text; a few examples of our changes include:

- Lines 144-147: We ran a sensitivity analysis with alternative definitions of *engage* for agriculture — definitions which accounted for the group's power to form a Groundwater Sustainability Agency via a governing entity such as a reclamation or irrigation district, which is not an option available to the other stakeholder groups (Supplementary Section 5).
- Lines 379-383: Agricultural users often are represented by agricultural entities within Groundwater Sustainability Agencies. These entities include reclamation and irrigation districts, or more broadly, entities with formal governance authority. Here, formal governance authority means that agricultural stakeholders can participate in the process on their own accord, i.e., without needing to be invited by others with power.
- Lines 396-404: In contrast to agriculture, opportunities for domestic and environmental groups to influence decision-making processes are limited despite the regulatory requirement to consider these users in Sustainability Plans. Often, stakeholder groups that lack formal governance authority to participate, such as domestic and environmental groups, also lack financial resources, time, and technical capacity to engage meaningfully in the planning process^{32,38}. Early stakeholder engagement in the policy processes is important, but it is also key to consider how and when to engage stakeholders throughout the entirety of the process³⁹. Without the ability to form a governing body to represent their interests, the most straightforward route for domestic and environmental stakeholders to influence decision-making processes is to be appointed stakeholder seats on governing boards.

We recognize the importance of literature in supporting our approach, and we now cite literature on power within the manuscript to justify this revised approach. Given the word count set by the journal for the manuscript, we expanded on this in a new section in our Supplementary Material (Supplementary Section 5. *Sensitivity Analysis for Agriculture Engage Component*), which allowed us to discuss the literature in greater detail. In this Supplementary Section, we expand on the theoretical justification for the distinction between Sustainability Plan governing board seats for invited stakeholders and Sustainability Plan governing board seats for a Groundwater Sustainability Agency, which agricultural representatives can occupy if a Groundwater Sustainability Agency is, or includes, a reclamation district or irrigation district. Drawing on the Power Cube framework (see manuscript Ref 37), we see the need for an invitation to participate in decision-making versus the ability to form a Groundwater Sustainability Agency and participate on a group's own accord as referring to different 'spaces' of power that stakeholders have access to: 'invited' versus 'closed'. In the definition of stakeholder engagement that we use in our analysis, as part of the broader concept of stakeholder integration, we are most interested in stakeholder's participation in 'invited' spaces: that is, the invitation for stakeholders to participate in an advisory committee or hold a designated stakeholder seat on the Sustainability Plan voting board. These 'invited' spaces are those in which "people (as users, citizens or beneficiaries) are invited to participate by various kinds of authorities" (see manuscript Ref³⁷, pg. 26) – here the authority is the Groundwater Sustainability Agency(s). We focus on these invited spaces as they are areas for stakeholders to voice needs, preferences and perspectives that may not be represented by a governing authority.

Alternatively, the Power Cube framework defines 'closed' spaces as ones in which "decisions are made by a set of actors behind closed doors, without any pretense of broadening the boundaries for inclusion" (see manuscript Ref 37, pg. 26). Groundwater Sustainability Agencies are 'closed' in the sense that only

entities defined in the Sustainable Groundwater Management Act legislation have the authority to become Groundwater Sustainability Agency(s) and use the governing powers vested in them to write and implement Sustainability Plans. Only agricultural stakeholders, and only those that are part of a reclamation or irrigation district, have access to the “closed” space of Groundwater Sustainability Agency formation and decision making.

Our methodology has delineated between ‘invited’ and ‘closed’ engagement by coding only stakeholder participation in ‘invited’ spaces. We agree that agriculture’s participation in the ‘closed’ space of Groundwater Sustainability Agency governance is likely influential on the Sustainability Plan outcomes, but if we group these together, we are unable to differentiate between the two different forms of engagement and power.

Third, to further address Reviewer #2’s concern in our comparison of agricultural stakeholders with environment and domestic stakeholders, who have access to different spaces of power, we added a new sensitivity analysis to our manuscript. To conduct these new analyses in direct response to the reviewer comments, we 1) coded all 108 of the Sustainability Plans to assess if the agricultural group was represented by either an irrigation district or reclamation district acting as, or as part of, a Groundwater Sustainability Agency; and 2) ran the stakeholder integration and protection analyses for these added dimensions of Agriculture.

We are grateful for the Reviewer #2’s comments, as these additional analyses provide new insightful results (detailed below) that strengthen the main findings of our manuscript. Below, we summarize the new methods and results.

1. Recoding *Engage* and *Aggregate* for Agriculture.

For agriculture only, in addition to the original *engage* component that accounts for a designated stakeholder representative seat on a voting board, we coded two variations for the *engage* component (defined as *engage2* and *engage3* in the Supplementary Table 5.1). Each variation in *engage* consists of two elements that are combined, as in the initial coding, which are 1) representation on a stakeholder advisory committee and 2) representation on the Sustainability Plan board. We did not revise or vary the coding for representation on an advisory committee and this element stays consistent in each of the variations in coding. As such, *engage* for agriculture has the following three variations:

- (a) **Engage**. This original variable for agriculture represents the invited power in a stakeholder specific voting board position for the Sustainability Plan. This coding is consistent with the *engage* coding for domestic and environment.
- (b) **Engage2**. This new variable (*engage2*) for agriculture represents the ‘closed’ power inherent in a Groundwater Sustainability Agency voting board seat. We coded whether Sustainability Plans had an irrigation district or a reclamation district acting as, or as a part of, a Groundwater Sustainability Agency and combined this with the previous coding of participation on an advisory board (*engage2*).
- (c) **Engage3**. This new variable (*engage3*) gives the Sustainability Plan credit for both forms of ‘invited’ and ‘closed’ voting board positions (note that we did not give a plan an extra point for having both types of seats for agriculture).

2. Analyzing integration for the *engage2* and *engage3*.

Stakeholder integration with the new Agriculture scores

After coding all 108 of the Sustainability Plans, we ran all of our integration and protection analyses with these two new iterations of *engage* for agriculture, as well as new *aggregate* integration scores that reflect these changes, *aggregate2* and *aggregate3*, respectively. We found that Sustainability Plans have both an agricultural stakeholder specific voting board position and

an irrigation or reclamation district acting as, or as a part of, a Groundwater Sustainability Agency in five cases (*engage3*). In 34 Sustainability Plans, an irrigation or reclamation district was acting as, or as a part of, a Groundwater Sustainability Agency, and there was NOT an agricultural stakeholder specific voting board position (*engage2*), and in 16 Sustainability Plans, the Groundwater Sustainability Plan board was comprised of an agricultural stakeholder specific voting board position and irrigation or reclamation districts were absent from the board (*engage1*). In 50 Sustainability Plans, agriculture has no representation (Figure 1).

Figure 1. Results from coding Agriculture representation on the Sustainability Plan governing board to account for both ‘invited’ and ‘closed’ spaces. The variations here (Stakeholder seat, Agriculture entity only, both, or neither) reflect the three variations of *engage* (*engage*, *engage2*, and *engage3*; respectively). Note that *engage* in each variation, also includes an assessment of if the Sustainability Plan has an advisory committee seat for agriculture, which stays constant between each variation.

Overall, it is more common to have an invited agricultural stakeholder board seat when agricultural interests are not already represented by a reclamation or irrigation district board seat, than to have both an agricultural entity (i.e., irrigation or reclamation agency) and a designated agricultural stakeholder serve on the board. When we compared the new *engage2* and *engage3* variables for agriculture with the initial coding of *engage*, we found a statistical difference between the three groups. With a post-hoc test, we identified that the difference was driven by the the scores for *engage2* and *engage3*, such that Sustainability Plans generally have higher scores for *engage3* than *engage2* (Supplementary Figure 5.1 and Supplementary Table 5.2). This was expected given that *engage3* is a combination of the original coding for *engage* and *engage2*. It is, however, surprising that the original coding for *engage* is not statistically different from both of the new variables.

Stakeholder integration with the new Agriculture scores—comparison to other stakeholders

When we compared the new agriculture variables with the *engage* scores for domestic and environment, we found that both *engage2* and *engage3* scores for agriculture were statistically higher than the domestic and environment *engage* scores (Supplementary Figure 5.1 and Supplementary Table 5.3). In the initial analysis, *engage* for agriculture was not statistically different from *engage* for environment. As the reviewer suggests, when we considered this additional form of engagement (i.e., agriculture representation via an irrigation or reclamation district on the board), we found evidence that agriculture is more engaged than environmental and domestic stakeholders. This aligns with previous research on the Sustainable Groundwater Management Act that has shown that agricultural interests are more well represented in GSA

boards than other users (see manuscript Refs 23, 26, & 36). Nevertheless, when we compared the new *aggregate* integration scores (*aggregate2* and *aggregate3*) with the *aggregate* integration scores for environment and domestic stakeholders, we saw no difference in the analyses compared to the initial coding of *aggregate* integration for agriculture. This is likely because *engage* is just one of the four components that we use to define integration.

3. Analyzing mixed-methods (integration - protection) outcomes for the *engage2* and *engage3*. Stakeholder protection with the new Agriculture scores

In addition to running the sensitivity analysis for integration, we ran the mixed methods models; this allowed us to assess the relationship between the new *engage2*, *engage3*, *aggregate2* and *aggregate3* variables for agriculture and Sustainability Plan percent coverage and protection for agricultural wells. This is in direct response to the reviewer's comments that "if agriculture did not have institutional representation, stakeholder integration per the authors definition may indeed have an impact."

By comparing the model results for our initial coding of *engage* and *aggregate* for agriculture with these new versions, we could understand what the different forms of power in engagement mean for well protection and coverage (i.e., *engage* and *aggregate* represent just 'invited', *engage2* and *aggregate2* represent just 'closed', and *engage3* and *aggregate3* represent 'invited' and 'closed'). For the linear models testing the relationship between *aggregate2* and *aggregate3* with agricultural protection and coverage (Supplementary Table 5.5 and Supplementary Figure 5.5), we saw no change in the model results in the new aggregate scores vs. covered and protected compared to the initial coding, i.e., there was no statistical relationship between *aggregate2* and *aggregate3* with neither agricultural well coverage nor protection. Again, this was likely because the shift to the new version of *engage2* and *engage3* had a relatively minor impact on the overall *aggregate2* and *aggregate3* scores respectively, as they also include the scores for *describe*, *analyze*, and *act*.

We did find a new result when we evaluated the linear models examining the relationship between agriculture's four integration components, including *engage2* and *engage3*, and agricultural protection and coverage (Supplementary Figure 5.5 and Supplementary Table 5.6-5.9). In the original models, we found that *engage*, *describe*, *analyze*, and *act* scores were non-significant for protection and coverage. This remains the same in the new models (note that the scores for *describe*, *analyze*, and *act* do not change), except for *engage2* in the model predicting agricultural coverage, which becomes a highly significant negative predictor of agricultural well coverage (Supplementary Table 5.6). As *engage2* represents agricultural access to 'closed' spaces as a reclamation or irrigation district Groundwater Sustainability Agency, this suggests that when agricultural interests are represented within Groundwater Sustainability Agencies, the boards were statistically more likely to select or place representative monitoring wells in places that were not close to agricultural wells.

We report on this result in the manuscript as potential evidence for the 'hidden' power of agriculture interests, as opposed to visible or observable power, using another dimension of power in the Power Cube framework (see Ref 37 in manuscript), to reduce the impact of Sustainability Plans on agricultural wells.

- See lines 385-394: The literature on power in decision-making processes³⁷ highlights pathways for "hidden" power dynamics, or ways in which some stakeholders may wield power in forms that may not be easily observable. For example, agricultural governing bodies can execute power in setting the Sustainability Plan agenda, deciding priorities and processes for Sustainability Plan development and deciding whether to invite other stakeholders in on advisory committees or board voting seats. In our sensitivity analysis

we found evidence for what may be agricultural governing bodies executing this less visible power: Sustainability Plans with agricultural governing bodies within Groundwater Sustainability Agencies are associated with less coverage for agriculture (Supplementary Section 5). Less coverage for agricultural wells means that there are fewer representative monitoring sites where agriculture is pumping groundwater, which could lead to undetected undesirable results.

Overall, these additional analyses allowed us to differentiate between ‘invited’ stakeholder participation and agriculture’s unique formal governance authority, i.e., ability to form or take part in a Groundwater Sustainability Agency, in our results. We still did not see that integration into the Sustainability Plan for agriculture is associated with protection of agricultural wells under any of the definitions for agricultural engagement; we suggest this is because agriculture’s influence is likely more hidden and not captured in our definition of integration. Nevertheless, we believe this further strengthens our initial findings that the ‘invited’ positions for environmental and domestic stakeholders matter, in large part because they do not have formal governance authority, i.e., access to these ‘closed’, more powerful positions.

Comment 3. I am also a bit unconvinced by the deployment of equity as a focal concept for the paper. Lines 80-82 provide the authors definition of equity as how similarly each user group is protected. The manuscript asserts that stakeholder integration enhances equity for all stakeholders, but the results clearly show that ecosystem protection lags far behind agriculture and domestic stakeholders. To me the results show improved outcomes for the stakeholder groups, not equity. If the authors wish to keep equity in the paper, I think there is need for more nuance and complexity in the discussion, especially since less disparities in protection levels between these groups doesn’t account for variable impacts of those levels as the authors discuss in the discussion.

We appreciate Reviewer #2’s call for a more nuanced discussion of equity in our deployment of the term within the paper. We agree with their assessment and, as a result, we de-centered equity as a concept in the paper. Moreover, in response to the reviewer’s comment, we (1) revised our title and terminology; (2) performed an additional analysis to further explore protection among stakeholder groups; and (3) highlighted the need for further research in this area. These changes are detailed below.

1. We revised the title of our paper to remove the focus on equity. It now reads: “Stakeholder integration predicts better outcomes from groundwater sustainability policy”. We removed discussion of equity from our introduction and results; given the word count set by the journal, we did not feel that we had the space to provide the nuances needed to adequately discuss and define the multiple dimensions of equity.
2. We conducted an additional analysis to test how similarly each user group is protected (i.e., formerly referred to as equity within the paper) by comparing the protection scores from those plans with the highest and lowest stakeholder integration *aggregate* score tertiles. We then ran a new analysis using one-tailed t-tests and analysis of variance tests to compare the mean protection scores for two groups: the lowest third of aggregate integration scores and the highest third of aggregate integration scores. We ran this both within and between stakeholder groups to test our assertion that the stakeholder high groups’ mean protection scores are more similar, e.g. reduced disparity, than the stakeholder low groups’ mean protection scores.

We find, similar to our original results, that the low vs. high group difference is significant for only the environment and domestic groups, and not agriculture; these results, again, suggest that integration is positively associated with protection for environment and domestic users, but not associated with protection for agricultural users.

We find that between groups, there is significant disparity in scores for environment, domestic and agriculture in the low group, such that environment’s mean protection score is significantly

lower than domestic's score, which are both significantly lower than agriculture's score. Among the high group, the disparity between agriculture's and domestic's mean protection score is greatly reduced and no longer statistically distinguishable, but the environment is still significantly lower. While the results are more "similar" between agriculture and domestic, they are not more similar for agriculture and environment or domestic and environment.

We include this analysis in our results section (**Stakeholder integration associated with protection**), with supporting material in Supplementary Tables 4.8, 4.9, 4.10 and Supplementary Figure 4.1; additionally, we added a new figure (Figure 4) to show these results.

3. In this revision, we took care to clarify the language used within our manuscript. For example, we do not perceive the outcomes as equitable – as the reviewer points out, disparity remains – and we took care to clarify this within the text.
 - See lines 406-420: Our results suggest that invited and visible stakeholder integration for domestic and environmental groups is a significant avenue towards improved outcomes for these stakeholder groups. This is not to say that it is sufficient to produce better outcomes for these user groups. In some cases, stakeholder engagement can be a form of token participation that gives the facade of diverse inclusion, but lacks an ability to sway hegemonic power relations or majority voting rules⁴⁰. For domestic stakeholders, higher levels of integration were associated with similar levels of protection to agricultural stakeholders. Although protection for environmental stakeholders increased with higher levels of integration, environment's protection still lagged domestic and agricultural stakeholders significantly. Surprisingly, we find no evidence that disadvantaged communities benefit from stakeholder integration, but this finding is likely because no plan had high integration scores for disadvantaged communities. Without examples of high integration for disadvantaged communities, we cannot statistically interpret this result to suggest that integration is not important for disadvantaged communities; based on the results from the domestic and environment groups, it is likely that integration is important for disadvantaged communities. More research is needed to understand the drivers and processes of equity in groundwater governance processes^{41,42}.

Comment 4. Relatedly, not including DACs in the analysis of the effects of integration feels like a major missed opportunity to more meaningfully get at equity in the paper as I discuss further in the specific comments below.

We thank Reviewer #2 for this suggestion and agree that a DAC integration assessment would be a great addition to the paper. We initially chose to exclude this analysis because we set up our spatial analysis as DAC wells versus non-DAC wells, which we could not replicate with our integration coding (i.e., we only had coded integration of DACs and couldn't differentiate them from non-DAC stakeholders). Regardless, this comment made clear that we could run the analysis using the DAC integration data with the coverage and protection statistics for *just* the DAC designated wells (i.e., we did not need to have a comparison with non-DACs).

In response to the reviewer's comments, we ran ordinary least squares models to assess if integration of DACs was predictive of coverage and protection for DACs. We found no significant relationships to report on.

- See lines 292-295: Using ordinary least-squares linear models, we found that stakeholder integration for disadvantaged communities was not predictive of coverage or protection. This was the case for the *aggregate* score and the individual integration components (Supplementary Tables 4.11-4.12).

We include information regarding this new analysis in multiple sections. To incorporate all of Reviewer #2's comments, we needed to revise the structure of our results. Therefore, we now devote a new subsection in the results, 'Economically vulnerable groups are less integrated and protected', to discussing these results. (Previously, our results related to DACs were integrated throughout the other results section and introduced after the stakeholder group results.)

Additionally, we include new discussion of these results. Overall, integration levels for DACs were very low when compared to the main stakeholder groups. We suggest that the absence of a relationship is likely related to the absence of high integration scores for DACs, and should not be taken as evidence that stakeholder integration is not important for DACs.

- See lines 413-420: Surprisingly, we find no evidence that disadvantaged communities benefit from stakeholder integration, but this finding is likely because no plan had high integration scores for disadvantaged communities. Without examples of high integration for disadvantaged communities, we cannot statistically interpret this result to suggest that integration is not important for disadvantaged communities; based on the results from the domestic and environment groups, it is likely that integration is important for disadvantaged communities. More research is needed to understand the drivers and processes of equity in groundwater governance processes^{41,42}.

Comment 5. Third, I have several questions and concerns about the stakeholder integration score. That environment has the highest aggregate score seems odd based on existing literature about SGMA. As you note this is driven by the high describe component score. This, in turn, I suspect is driven by the requirement to map groundwater dependent ecosystems noted in supplementary table 2.1. If the other stakeholders don't have similar, specific requirement, I suspect this finding is picking up on this requirement (which based on the results seems relatively ineffective) rather than reflecting stakeholder integration and really confounds interpretation. The extent to which similar statutory requirements could be affecting other components should also be considered.

We thank Reviewer #2 for their deep engagement with the manuscript and SI; the reviewer's comments pushed us to better clarify our methods and approach.

We created Supplementary Table 2.1 explicitly to address the concern raised here and to be transparent in the regulatory requirements related to the different components of stakeholder integration into the Sustainability Plan. As noted in Supplementary Table 2.1 there are two different requirements for description relating to groundwater dependent ecosystems and agriculture and domestic stakeholders, as represented by primary use or uses for each aquifer. Both require the identification of the stakeholder groups, and we acknowledge that the requirement for groundwater dependent ecosystems is more explicit in the legislation, even if both sections generally request the same information for the stakeholder groups.

We agree with Reviewer #2 that this is likely driving the higher integration score for the environment.

- See lines 306-320: In the name of local control and flexibility, California's Sustainable Groundwater Management Act is a state-mandated participatory process¹⁹ that gives discretion to

local Groundwater Sustainability Agencies in how they integrate stakeholders. Drawing from the literature on collaborative governance and planning theory, planning processes are described as tiered stages through which a problem is defined, approaches to addressing the problem are identified, and then a decision is made to select the approach(es)^{9,20}. As such, we expected that the *engage*, *describe*, *analyze*, and *act* components would progress sequentially, representing coordinated integration of local knowledge and social learning through stakeholder participation^{8,12}. Instead, our results suggest that stakeholder integration in the majority of plans was not a sequential, coordinated process. Although domestic and environmental stakeholders are well described and considered with respect to potential groundwater loss (i.e., through high *describe* and *analyze* scores), their needs are not addressed in plan actions (i.e., explicit benefits from management actions captured in *act* scores). Alternatively, agricultural stakeholders achieve the same level of support through plan actions (*act*) as the environmental stakeholders, regardless of agriculture's *engage*, *describe*, and *analyze* scores. Our results likely reflect varied incentives and motivations to address stakeholder needs, and could be the result of regulatory requirements (Supplementary Table 2.1), local dynamics, or both.

We posit that the lack of clarity surrounding our definition of integration has also contributed to the concerns laid out by the reviewer; therefore, we clarified our definition and operationalization of stakeholder integration.

- See lines 81-87: We define integration as how well each Sustainability Plan incorporates stakeholders, their knowledge, and needs across four components: *engage*, *describe*, *analyze*, and *act* (Fig. 1b). We use the four components as a proxy for stakeholder integration into the local policy processes that resulted in each of the Sustainability Plans; the plans reflect only the stated and visible priorities of the Groundwater Sustainability Agencies, and not invisible power dynamics that determine decision-making powers within these local agencies.

We define integration as the integration into the Sustainability Plan and hypothesized that the components of integration would build upon each other. We see this result as deepening our understanding of stakeholder integration in that we can see that the high score for *describe* for environment is likely driven by a regulatory requirement, but that it doesn't necessarily correlate with higher levels of *analyze* and *act* as we had hypothesized it would. This helps us to see that these elements, in many cases, are operating independently and due to different State directives, incentives, and motivations, in this case likely due to the more explicit wording for description for groundwater dependent ecosystems than agricultural or domestic stakeholders. Importantly, it is the Sustainability Plans where *aggregate* integration scores (i.e., across the four components) are high that we see an association with protection.

- See lines 423-425: Overall, for the majority of Sustainability Plans, stakeholder integration is not a sequential, coordinated process, but for the few Sustainability Plans that integrate comprehensively, domestic and environmental stakeholders are more protected.

Comment 6. Related to the “act” component, I have a lot of questions about what was counted for the scoring here and need more information to interpret this element effectively. However, as described, this component feels more akin to an outcome or output rather than stakeholder integration. This needs to be defended as it is odd to use an outcome to predict another outcome. Would we not expect that the two to be correlated?

We appreciate the reviewer drawing attention to our definition of integration and its components, particularly the *act* component; in response, we took the opportunity to add language and be more transparent in our approach and methods. We feel that this comment relates again to the need to further clarify our definition of integration to clarify that we are analyzing stakeholder integration into the Sustainability Plan (see responses above for how we addressed increasing the clarity surrounding our use of integration).

For each stakeholder integration component, we are looking to the formal and explicit representation of the stakeholder in the Sustainability Plan. As the plan is our unit of analysis, the inclusion of projects and management actions to support a user group, as codified in *act*, represents a component of integration and not an outcome. Stated projects and management actions represent a priority or intention to act, such that *act* represents a form of support for the stakeholder group – a stated intention to support the user through a project or management action, theoretically in response to an identified need. We are aware that other formations of research questions and analyses of management plans could posit that the selection of projects and management actions are outcomes (and we are aware of at least one SGMA paper that does this, see manuscript Ref. 41).

With regard to the scoring of the *act* component, we revised the language in Fig. 1 to clarify that *act* captures if a stakeholder group is identified as a targeted beneficiary of a project or management action. For example, if a project or management action was intended to benefit all users, we did not code this as benefitting one of our stakeholder groups. Only when the project identified a stakeholder group by name, for example domestic users, did we code it as affirmative for *act*.

- See Fig. 1: Act - Stakeholders are identified as targeted beneficiaries of Sustainability Plan project and management actions.

We also reflect on the operational definition of *act* more explicitly in the discussion.

- See lines 314-317: Although domestic and environmental stakeholders are well described and considered with respect to potential groundwater loss (i.e., through high *describe* and *analyze* scores), their needs are not addressed in plan actions (i.e., explicit benefits from management actions captured in *act* scores).

Comment 7. Lastly, I worry that unobserved variables are driving the OLS results. The fact that no single component explains the aggregate score effect highlights this potential for me. I am sympathetic to the data challenges here but where possible it would be great to try out a few controls, at least maybe the total number of individuals per plan for the category and basin prioritization status. At the very least, the manuscript should include a discussion of this limitation and others.

We thank Reviewer #2 for bringing up this important consideration for our methods. It is a possibility that unobserved variables could be contributing to the ordinary least squares model results, and we added a limitations section to our manuscript to note this as a possibility.

- See lines 707-716: Sustainability Plans reflect only the stated and visible priorities of the Groundwater Sustainability Agencies, and therefore, are not representative of all of the interactions and power dynamics within the policy process. As such, omitted variable bias was a potential challenge for our models. Local-level, observational variables that were not formalized in the Sustainability Plan were outside the scope of our study. Given the sample size and lack of hypothesized alternative predictors, we focused our study on stakeholder integration and its influence on stakeholder outcomes (i.e., coverage and protection). Nevertheless, we ran additional post-hoc exploratory analyses, using t-tests and ordinary least squares models, to examine whether stakeholder integration or outcomes varied with certain variables, such as the submission year of the Sustainability Plan (2020 versus 2022) and the number of individuals (i.e., total number of wells or ecosystem area) per plan (Supplementary Section 6).

Given that this is a mixed methods approach, where we quantitatively coded results from Sustainability Plans, it is possible that other variables that are not observable in the plans, such as participant trust in each other, legitimacy of governing bodies, financial capacity for outreach and engagement, are related to our integration scores and are contributing to the ordinary least squares model results. In fact, we anticipate that there is a whole suite of variables at play that are commonly examined via case study approaches. Given the scope of our study, evaluating 108 Sustainability Plans, these local-level,

observational variables that are not formalized in every plan are outside the scope of our study. We readily acknowledge this limitation. We, however, believe that the scale of our study is a valuable addition to the literature.

We acknowledge that the reviewer has requested that we incorporate controls into our ordinary least squares models in an effort to control for the influence of omitted variables. Given our sample size (n=108) – which is large for comparing between policy processes, but small for statistical power – and the hypothesis being tested in our models – that the integration components of the Sustainability Plan relate to coverage and protection – we were strategic about the relationships that we tested so as to not undermine the statistical power of our models. We considered adding control variables at various points in our analysis (pre-submission, as well as during this revision), but could not identify controls that represent clear, evidence-based relationships with our independent and dependent variables.

Both of the variables suggested by the reviewer – number of stakeholder individuals in a plan and basin prioritization status – represent ambiguous constructs when considered in relation to stakeholder integration and stakeholder protection and coverage that confound the interpretation of our models:

- **Basin prioritization status.** In California, groundwater basins are prioritized as critical, high, medium, low and very low according to a ranking scheme based on data such as population and water usage estimates (<https://water.ca.gov/Programs/Groundwater-Management/Basin-Prioritization>). Under the Sustainable Groundwater Management Act, basins that were given a status of critical were the first plans required to submit their plans in 2020, whereas the remaining high and medium priority basins were required to submit their plans two years later, in 2022. Using basin prioritization status as a variable confounds the severity of the groundwater problem with the amount of time available to each basin to complete their Sustainability Plan. As such, we can not differentiate between the severity of groundwater problems and amount of time to complete a Sustainability Plan. Regardless, we do not have any reason to anticipate that any of these elements would undermine our characterization of integration or protection or coverage. We do acknowledge that understanding how integration, protection and coverage vary by year of submission is useful. As such, we added an additional analysis in the SI, which we reference in the manuscript, to explore via two-sided t-tests and Kruskal Wallis tests, how integration, coverage and protection vary by submission year (Supplementary Tables 6.2 and 6.4).

Overall, we see that integration scores are significantly higher in the 2022 plans than in the 2020 plans. We also see that most stakeholder groups had significantly higher coverage and protection in the 2022 plans than in the 2020 plans (except for agricultural protection which had no significant difference between the years).

- **Total number of stakeholder individuals per plan.** The number of stakeholder individuals in a Sustainability Plan could influence integration, coverage, and protection in multiple ways, such that we feel inclusion in the model would not control or clarify the relationship between integration and coverage and protection. For example, increasing population in the plan area for a given stakeholder group could suggest that there is a large interest group with motivation and funding to act, or we could hypothesize that a larger stakeholder presence would decrease the imperative of individuals to act and advocate in a crowded field. Nevertheless, as an exploratory analysis, rather than as a control variable, we see this as a valuable relationship to explore. We added an additional analysis using ordinary least squares models in the SI to explore the relationship between the total number of agricultural wells, domestic wells, and acres of groundwater dependent ecosystems in each plan area with respective coverage and protection (Supplementary Table 6.3) and stakeholder integration (Supplementary Table 6.4). This analysis has yielded an interesting finding: we found for each stakeholder group, the number of wells or

acres is significantly and negatively associated with coverage, suggesting that as the population of wells and acres grows, the Sustainability Plan's representative monitoring systems do not scale to cover the population. We find no significant relationships between protection and population, nor with the *aggregate* integration score and population. Given the scope of our analysis and the word limits of the journal, we are not reporting on these results in our manuscript, but we do include them in Supplementary Section 6.

In sum, the addition of a limitations section into the main manuscript and the additional exploratory analyses for year of plan submission and stakeholder population has improved the manuscript and has allowed us to present our results more transparently. Thank you.

Specific comments and suggestions

Comment 8. Abstract line 38: This sentence feels overly specific considering existing literature. I agree that the literature theorizes that integrating stakeholders can provide protections from adverse management impacts but I don't think there is specific literature to this point about groundwater depletions nor do I agree that groundwater depletion is an adverse impact when it comes to all stakeholders. I suggest re-writing in a more general way.

Agreed, the abstract has been edited.

- See lines 37-39: Natural resources policies that promote sustainable management are critical for protecting diverse stakeholders against depletion. Although integrating diverse stakeholders into these policies has been theorized to improve protection, empirical evidence is lacking.

Comment 9. Abstract missing two of three key findings as defined in the manuscript (lines 101-112)

The abstract now contains all of our key findings summarized.

- See lines 41-44: We find that the majority of Sustainability Plans do not integrate or protect the majority of their stakeholders. Nevertheless, our results show that when stakeholders are more integrated into a Sustainability Plan, they are more likely to be protected, particularly for those that lack formal access to decision-making.

Comment 10. I really appreciate the clear and effective summation of findings at the end of the introduction.

Thank you.

Comment 11. Lines 114-124: Really can't tell from this paragraph what those four components mean. For transparency these components and their scoring should be described in the manuscript not just the supplemental.

We appreciate this comment and agree that we needed more clarity in the presentation of the integration components.

- See lines 81-87: We define integration as how well each Sustainability Plan incorporates stakeholders, their knowledge, and needs across four components: *engage*, *describe*, *analyze*, and *act* (Fig. 1b). We use the four components as a proxy for stakeholder integration into the local policy processes that resulted in each of the Sustainability Plans; the plans reflect only the stated and visible priorities of the Groundwater Sustainability Agencies, and not invisible power dynamics that determine decision-making powers within these local agencies.
- See lines 119-125: More specifically, we defined four generalizable integration components (Fig. 1b), for which we asked, how well did each Sustainability Plan: *engage*, *describe*, *analyze* impacts of depletion on, and *act* to support each stakeholder group and subgroup as evidenced within the text? Our rubric enabled us to calculate scores (ranging from zero to two) for each integration component, and to sum integration components to make an *aggregate* score (ranging

from zero to eight); the *aggregate* score assessed how well each Sustainability Plan integrated each stakeholder group across the four components (Supplementary Fig. 2.3).

- See Fig. 1b, which has updated definitions.

Comment 12.

Lines 135- 138: Please clarify what this means.

For reference, lines 135-138 in the original document: Comparing agriculture, domestic and environment stakeholders via Kruskal-Wallis tests and pairwise Dunn tests we saw no clear pattern arise between stakeholder scores across all integration components, despite significant ($p \leq 0.01$) differences between groups. We found agriculture scored higher for the engage ($p \leq 0.001$) and act ($p \leq 0.001$) components but lower for describe ($p \leq 0.01$) and analyze ($p \leq 0.01$) than domestic.

Thank you. We edited this paragraph to increase the clarity of our objective and the results.

- See lines 132-135: Comparing agricultural, domestic, and environmental stakeholders via Kruskal-Wallis tests and pairwise Dunn tests, we saw no overall pattern between stakeholder integration component scores — no stakeholder group scored consistently higher or lower than other groups across components — despite significant ($p \leq 0.01$) differences existing for specific components between groups.

Comment 13. Line 199: “as horizontal distances...” Really important to be super clear here and elsewhere this is a methodological/sensitivity analysis point. The material impact on “uncovered” wells is not changed. Again on line 280 as well.

We appreciate the Reviewer #2’s suggestion to add clarity; we adjusted the wording.

- See lines 154-166: We analyzed which stakeholder groups were “covered” and “protected” by each Sustainability Plan’s groundwater level minimum thresholds established at representative monitoring wells (Fig. 3). Minimum thresholds measure where undesirable results (e.g., a well running dry or an ecosystem die-off) may occur if groundwater levels decline. For each instance of a stakeholder (i.e., domestic well, agricultural well, or groundwater-dependent ecosystem), we identified monitoring wells that were nearby: at or within a horizontal distance of 2.4 km (1.5 miles; Fig. 1c-d). Stakeholders nearby monitoring wells were considered covered and were assessed for how well the Sustainability Plan protected the stakeholder from losing access to water as a result of declining groundwater levels. Individual stakeholders not covered by a monitoring well were deemed not protected. Protection was determined by comparing each stakeholder’s groundwater access depth (e.g., well depth or maximum rooting depth for vegetation) to the minimum threshold groundwater level established at the nearby monitoring well. If a stakeholder’s groundwater access depth was equal to or shallower than the minimum threshold at the nearby monitoring well, the stakeholder was not protected (Fig. 1c-d).
- See lines 177-184: We ran a sensitivity analysis, adjusting the horizontal distance surrounding each monitoring well — 0.8 km to 4.0 km — to assess the impact on coverage and protection (Supplementary Table 3.1, Supplementary Figs. 3.2-3.7). As the horizontal distance increased, the proportion of wells and ecosystems covered and protected within each Plan increased. The increases in protection did not scale proportionally across stakeholders. For example, using the largest horizontal distance (4.0 km), 78%, 77%, and 61% of agricultural wells, domestic wells, and ecosystems were covered, but only 64%, 59%, and 15% were protected, respectively. In short, the disparity in protection among agricultural, domestic, and environmental stakeholders is exacerbated as coverage increases.

Comment 14. Lines 204-206: I think this is backwards. As I read SGMA, the intent was to define significant and unreasonable and then define MTs to avoid those conditions.

We agree. We appreciate the opportunity to be more precise in our language. We changed the sentence.

- See lines 186-194: In addition to quantifying protection, we performed a review of each Sustainability Plan’s “significant and unreasonable” conditions to examine if and how these are established to protect stakeholders (Cal. Water Code § 10721(x), 23 Cal. Code of Regs. §§ 354.26, 354.28). Groundwater Sustainability Agencies are given the discretion to define minimum thresholds in order to avoid “significant and unreasonable” conditions. Twenty-four percent of Sustainability Plans (n=26) established minimum thresholds specifically to be protective of agricultural wells, domestic wells, groundwater-dependent ecosystems, or a combination of stakeholders. The remaining 76% used non-stakeholder related approaches to define minimum thresholds, such as lowest historical well levels (Supplementary Section 3; Supplementary Tables 3.6-3.7).

Comment 15. Lines 207-209: Can you provide an example of what that looks like. Is it meaningful if not reflected in MTs? As written, it isn’t clear to me what this analysis of sustainability goals relates to your research question. I recommend putting this info in conversation with the other findings or delete.

We significantly revised this paragraph to clarify the definition of minimum thresholds and put the results in conversation with the protection and coverage results (see comment 14 response).

Comment 16.

Line 215-218: Lines 148-157 you talk about measuring integration for these subgroups and you have spatial data on the DACs so I don’t understand why DACs are not included. What does including these subgroups in the paper accomplish if they are not part of this analysis? For me, this really limits the equity discussion.

Please see response to Comment 4, where we outline a new analysis (i.e., ordinary least squares models that assess if integration of disadvantaged communities was predictive of coverage and protection for disadvantaged communities) that addresses both comment 4 and this comment.

Comment 17. Lines 229-240: I don’t feel like this paragraph does as good of a job as it could reflecting the nuances of the results from Fig 4. Would like to see a bit more clarity parsing out the individual score component results.

We clarified the findings and we added text to the discussion to provide additional context to the results and highlight why they matter. Please note that Fig 4 is now Fig. 5 in the revised manuscript.

- See lines 236-247: Additionally, we assessed if each component (i.e., *engage*, *describe*, *analyze*, *act*) independently predicted stakeholder coverage and protection using ordinary least-squares linear models (Fig. 5g-l; Supplementary Tables 4.2-4.7). Coverage and protection for agricultural stakeholders, as well as coverage for domestic stakeholders, were not significantly influenced by *individual integration* component scores. Nevertheless, for domestic stakeholders, a high score for *analyze* and *act*, compared to a low score, was associated with 15% ($\beta=7.47$, $p\leq 0.05$) and 16% greater well protection ($\beta=8.19$, $p\leq 0.01$), respectively. For environmental stakeholders, a high score for *engage*, compared to a low score, was associated with 26% greater coverage ($\beta=12.97$, $p\leq 0.01$) and 12% greater protection ($\beta=6.30$, $p\leq 0.05$). Additionally, a high score for *analysis*, compared to a low score, for environmental stakeholders was associated with 16% greater protection ($\beta=7.86$, $p\leq 0.05$). In short, no integration components are associated with higher coverage or protection for agricultural stakeholders, and no single integration component consistently explains the variation in coverage and protection for both domestic and environmental stakeholders.

Comment 18. In Supplementary table 2.1 “describe” “response levels” What do X/Y and Z mean?

We updated Supplementary Table 2.1 to better define the *describe* component to clarify that we are referring to latitude, longitude and depth.

Comment 19. Did you do any analysis of potential differences based on basin prioritization? Feasibly with less time to write plans there may be a different effect of stakeholder integration.

Please see response above to comment 7 where we describe our new SI analysis regarding the year that Sustainability Plans (related to basin prioritization) were submitted and our discussion of basin prioritization.

Comment 20. I think it would be very helpful to add examples to the supplementary table 2.1 of the response levels for greater transparency/clarity.

We have added additional text to Supplementary Table 2.1 to provide more clarity in the response levels.

Comment 21. Line 249: clarify here or on lines 125-127 previously if comprehensively means getting at least a one on each of the four or scoring a 2 on each

We clarified this in the results (i.e., we edited the text on lines 125-127 of the original document).

- See lines 126-127: Most Sustainability Plans failed to comprehensively integrate stakeholders: only 9% of plans achieved a score >0 for all components for all stakeholder groups.

Comment 22. Lines 269-271 – Why does this specifically apply to marginalized users and not the three stakeholder categories broadly? I would argue this is why you see such low integration period not just for those two subgroups

We agree and clarified this in the text.

- See lines 322-338: Although some stakeholder groups excelled at some components, we found no compelling evidence that any stakeholder group is more explicitly integrated in aggregate than another (this includes when we account for agricultural stakeholders' differential ability to be represented by governing entities, such as reclamation and irrigation districts, on Groundwater Sustainability Agencies boards). Low integration of all stakeholders into plans is likely related to California's Sustainable Groundwater Management Act's regulatory language²¹ which prescribes processes for local entities to define and establish sustainable management criteria (Supplementary Section 1) without mandating protection outcomes specific to stakeholder groups. Alternate models of stakeholder-related provisions in natural resources regulation show that performance-based regulation (i.e., specifying the outcome to be achieved, rather than how to achieve it) in this area is possible: e.g., developing water plans that must not negatively impact the current levels of protection for specified stakeholder groups (Basin Plan 2012 (Austl.) section 10.54). Such an alternative model could be especially useful for the vulnerable subgroups. Although no single stakeholder group is more integrated than another, our results indicate that the two subgroups — small farms and disadvantaged communities — are less integrated than their respective stakeholder groups. Local-scale governance can incorporate resource users' preferences and valuable local knowledge²², but when directive is low and discretion is high, there is a greater risk that Sustainability Plans may be co-opted by economically powerful local interests at the expense of diverse voices^{20,23-27}.

Comment 23. Lines 278-290: I'm not clear how you get from there not being different levels of coverage and protection at the beginning of this paragraph to not protecting domestic and environment groups at the end. Implicitly I wonder if you are getting at the fact that the material impact to these users is likely more severe than to ag (a well-resourced farmer could just drill another well maybe) but this is not clear until the next paragraph. Needs refinement

We reorganized our discussion to clarify that these are two separate results that we contend are related - one relating to levels of coverage and protection for stakeholder groups and the other reflecting on the definitional criteria used by Sustainability Plans to define minimum thresholds. Please see the Discussion subsection, *Protection is unequal and vulnerable groups experience greater burdens*, for the updated text.

Comment 24. Lines 325-236: But regulatory decision-making power is included in your stakeholder integration rubric and scoring criteria under “engage” no? And it is feasible for domestic wells to have regulatory decision-making authority if they were given an assigned seat on a board? I think here and elsewhere you are blurring this concept with institutional representation.

Please see our response to Comment 2 above.

Comment 25. Line 334-335: What does “integrate” mean here? That the components are correlated?

We significantly revised this section and no longer include this sentence or the term “integrate” in this way.

- See lines 423-434: Overall, for the majority of Sustainability Plans, stakeholder integration is not a sequential, coordinated process, but for the few Sustainability Plans that integrate comprehensively, domestic and environmental stakeholders are more protected. Previous literature emphasizes that successful stakeholder engagement is built upon a foundation of fairness, trust, respect, and co-learning^{19,39}. Our approach reflects on these aspects of integration into Sustainability Plans as a proxy for stakeholder integration into the local policy processes that resulted in the 108 Sustainability Plans by evaluating four components: *engagement*, *description*, *analysis*, *action*. The stakeholder integration literature has focused a great deal on the specifics of the policy process that lead to better integration¹¹; we build on this here to demonstrate that when integration happens, it leads to better outcomes for stakeholders that lack formal governance authority. In short, when the four integration components are taken together, the needs and knowledge of stakeholders can become better represented into planning and policy outcomes, which can lead to better outcomes among diverse user groups.

Comment 26. For Figure 1 I am confused by panel b. What does “Yes” and “No” mean? Needs clarification.

We revised Figure 1 and removed the sub-groups from panel b to focus on the main three stakeholder groups.

Comment 27. Line 459: Five questions but four components? Please explain/clarify

We clarified in the methods that one component, *engage*, draws on two binary questions that are ultimately combined to create a three-level response. The remaining three components are each structured as a three-level response in the question.

- See lines 551-557: The final evaluation rubric consisted of five questions: two binary *engage* questions for presence of a designated stakeholder seat on an advisory committee and voting board that were combined for the score, and a single question each for *describe*, *analyze*, and *act*. These questions were evaluated for all three stakeholder groups (i.e., domestic, agriculture, environment) and both subgroups (i.e., disadvantaged communities, small farms) (Supplementary Table 2.1). Each component featured a three-level ordinal numerical response spectrum (i.e., no: 0; somewhat: 1; yes: 2).

Comment 28. Line 476: What is step three refined coding? Not clear.

We added language to clarify that the third step of coding was an additional coding of the plans that featured our final, refined set of integration questions.

- See lines 568-572: In total, there were three rounds of data collection (coding): (1) draft 2022 Sustainability Plans the Groundwater Sustainability Agencies released during May-December 2021 for public comment; (2) final 2022 Sustainability Plans submitted to the California Department of Water Resources in January 2022; and (3) re-coding of the final 2020 and final 2022 Sustainability Plans with our final, refined set of integration questions.

Comment 29. Are all wells in OSWCR included in the analysis or did you use an estimated retirement age to remove older wells? Needs to be clarified and the implications for the results should be discussed.

We added additional information to the methods to address this comment.

- See lines 607-616: We selected wells with a recorded construction date 1975-2022 to account for the retirement of older wells. We selected wells with the record type "WellCompletion/New/Production or Monitoring/NA", which best represents the construction of new wells. We selected only records with a reasonable depth (i.e., depth >0) and reasonable location (i.e., within CA). For domestic wells, we selected wells with "Water Supply Domestic" as the recorded use; for agricultural wells, we selected wells with "Water Supply Irrigation - Agriculture", "Water Supply Irrigation - Agricultural", "Water Supply Irrigation Agricultural", and "Water Supply Irrigation Stock or Animal Watering" as the recorded use. The database includes duplicate records; we selected records with distinct entries for well record number, latitude, longitude, well depth, construction date, and use.

Comment 30. Line 516: It is immediately clear to me why a normal distribution is the deciding factor here? Please clarify this rationale. Does the Department of Water Resources have any guidance on how much spatial area a monitoring well can reasonably represent?

The Department of Water Resources states in its Monitoring Networks and Identification of Data Gaps Best Management Practice Guidance Document (https://water.ca.gov/-/media/DWR-Website/Web-Pages/Programs/Groundwater-Management/Sustainable-Groundwater-Management/Best-Management-Practices-and-Guidance-Documents/Files/BMP-2-Monitoring-Networks-and-Identification-of-Data-Gaps_ay_19.pdf) that "there is no definitive rule for the density of groundwater monitoring points needed in a basin". Instead, the guidance document defers to professional judgment. Because groundwater conditions can be spatially heterogeneous within a groundwater basin, particularly near surface water features that have been subject to fluvial morphology, monitoring wells need to be sufficiently close enough to production wells (agriculture and domestic) and ecosystems to monitor groundwater conditions for that user. We selected a range of buffer distances, between 0.8 km (0.5 mi) and 4.0 km (2.5 mi), that are generally considered by hydrologic professionals and researchers to be a reasonable distance from a monitoring well for detecting groundwater conditions for a particular user. This range of buffers enabled us to then perform a sensitivity analysis of how the results changed according to the buffer distance. The normal distribution that we observed in the 2.4 km (1.5 mi; Supplementary Fig. 3.1) buffer tells us that monitoring well coverage in the basin will be more statistically representative across groundwater basins and less prone to skewed results.

- See lines 639-640: The 2.4 km buffer was the closest to a normal distribution and was selected for the analysis, because this buffer was least prone to skewed results (Supplementary Section 3).

Comment 31. Lines 128-130: I think it might be better to cite a specific paper you are following here rather than the broader literature. My mind immediately goes to Arnstein’s ladder of participation here, but I think the order of these steps would be different if you were following that model. To me engage is a much “more” of an ask than describe or analyze.

We updated this language in the text to clarify that we are pulling from multiple theories of the policy and planning process stages, specifically Ansell & Gash’s (see manuscript Ref. 9) description of collaborative governance processes and Escobedo Garcia and Ulibarri’s (see manuscript Ref. 25) description of planning processes.

- See lines 308-314: Drawing from the literature on collaborative governance and planning theory, planning processes are described as tiered stages through which a problem is defined, approaches to addressing the problem are identified, and then a decision is made to select the approach(es)^{9,20}. As such, we expected that the *engage*, *describe*, *analyze*, and *act* components would progress sequentially, representing coordinated integration of local knowledge and social learning through stakeholder participation^{8,12}. Instead, our results suggest that stakeholder integration in the majority of plans was not a sequential, coordinated process.

REVIEWER #3 (Remarks to the Author)

The analysis and findings in this paper offer the field of groundwater management, and natural resource management more broadly, useful quantitative evidence for the benefits of stakeholder integration. As the authors have stated in the manuscript, I am not aware of any other peer-reviewed research that has robustly quantified the impact of stakeholder integration on concrete resource management decision-making. Therefore, these findings are novel as well as useful for practitioners, policymakers, and other researchers.

We thank Reviewer #3 for their review.

The approach used to quantify stakeholder integration within the Groundwater Sustainability Plans is robust and well documented. The statistical analysis of the dataset is also well done with no major or minor flaws identified.

We hope that Reviewer #3 will also find the new analyses (in response to Reviewer #2) well done, well documented, and robust.

The discussion is helpful in addressing the differences identified between agriculture, domestic, and environment. The conclusions do a good job of articulating some of the broader connections that the research has beyond groundwater management in California.

We appreciate that Reviewer #3 found the discussion helpful and the conclusions articulate.

The only suggestions added, in the attached manuscript, pertain to some wording change suggestions and requests for clarity in the choice of terms in a few locations.

We thank Reviewer #3 for their encouraging remarks. Below, we copied their specific suggestions from the manuscript file so that we could explicitly highlight how we addressed each of their comments.

It may also be worth reviewing literature from the Colorado basin roundtable process for quantitative assessments of stakeholder integration on water management outcomes (if it wasn't reviewed already). Finally, it could be useful to note any additional research questions that the authors believe need to be answered or addressed next.

We thank Reviewer #3 for this suggestion. Below we summarize our review of the literature from the Colorado basin roundtable process. We were explicitly looking for quantitative assessments of stakeholder integration on water management outcomes; our exploration of the following five reports did not reveal any quantitative assessments of stakeholder integration.

1. Colorado Basin Needs Assessment Report.
<https://dnrweblink.state.co.us/CWCB/0/edoc/152957/ColoradoBasinNeedsAssessmentReport.pdf?searchid=dc0f7869-2787-49b9-9d2f-41261243aee4>
 - Acknowledges meeting needs of stakeholders and proposes projects/management actions to address them, but *no quantitative analysis of impact of stakeholder engagement*
2. Colorado Implementation Plan 2022 Update: Volume 2.
https://dnrweblink.state.co.us/CWCB/0/edoc/216708/Colorado_BIP_Volume2_2022.pdf?searchid=7e047589-d6f9-41d3-a865-f7e2917f6a2c
 - Acknowledges needs of stakeholders, includes the following language (page 2): “In response, Governor Hickenlooper issued an Executive Order (EO) in 2013 calling for the Colorado Water Conservation Board (CWCB) to work with the nine Basin Roundtables, the Interbasin Compact Committee (IBCC), and *other stakeholders to develop Colorado’s first Water Plan.*”
 - References to stakeholders participating in development of Integrated Water Management Plans, identifying data gaps and gap-filling approaches, and updating the basin Implementation Plan, but *no quantitative analysis of impact of stakeholder participation*

3. Colorado Basin Roundtable Nonconsumptive Needs Assessment Subcommittee Meeting Summary Nonconsumptive Needs Quantification Water Supply Reserve Account Grant Project October 19, 2009 and November 5, 2009 Meetings.
<https://dnrweblink.state.co.us/CWCB/0/edoc/138670/11-5%20Meeting%20Summary%20-%20CO%20Basin%20WFET%20Nov%202009%20FINAL.pdf?searchid=89c6f396-f95b-4d11-8a66-7dbc693303fb>
 - *No reference to stakeholder engagement* beyond statement identifying means of notifying stakeholder “[...] the group discussed how to gather input from local stakeholders [...] One of the first areas of gathering information from local stakeholders is to develop node maps for each SEO water district that the committee can then use for outreach in each water district.” (page 2)
4. Local Expert Procurement Process, Plans and Templates for Basin Implementation Plan Updates - January 2020. <https://www.coloradobasinroundtable.org/wp-content/uploads/2020/01/Procurement-Process-and-Roles-for-Local-Experts-1-15-2020.pdf>
 - *No reference to stakeholder engagement* except identifying “coordinat[ing] conversations between internal and external stakeholders to identify opportunities for improvement and areas where value can be added” as a ‘characteristic of the local expert’ (page 3)
5. Colorado River Basin Ten Tribes Partnership Tribal Water Study. Chapters 4 and 6.
<https://www.usbr.gov/lc/region/programs/crbstudy/tws/docs/Ch.%204%20Methodology%20of%20Assessments%2012-13-2018.pdf>
<https://www.usbr.gov/lc/region/programs/crbstudy/tws/docs/Ch.%206%20Colorado%20River%20System%20Effects%2012-13-2018.pdf>
 - This is the most in-depth discussion of stakeholder participation and the results of that participation: 4 scenarios were developed to project tribal water use/development. “Throughout the scenario planning process, the *Partnership Tribes were substantially involved* in determining the factors that influence future tribal water development. The scenario planning process and its outcomes reflect the perspectives that the *Partnership Tribes determined are critical to their future water development*” (Ch. 4, pg. 4-2).

We are very interested in this literature, so if there are additional roundtable reports that we overlooked, we would be interested in learning about them.

Specific comments copied from submitted word document

Lines 64-65: This is a true statement, at least as far as I know of in the academic literature, however, I wonder if there isn't grey literature that shows empirical evidence that this is true. What about work done by the River Network, WaterNow Alliance, US Water Alliance, or others? The Colorado Basin Roundtables may be another example of decentralized water management that has provided stakeholder integration, leading to more equitable outcomes. But I do not know if it's been studied or quantified.

We explored the gray literature, reviewing reports from the River Network, WaterNow Alliance, and US Water Alliance, as well as a dissertation. In our review of the gray literature, we did not find any reports or papers that show empirical evidence; most documents are about the development process rather than assessments of results of implemented plans after the fact. We did find and cite a report from a transdisciplinary group of dozens of scholars that took part in a series of workshops around the theme of “Advancing Scholarship and Practice of Stakeholder Engagement in Working Landscapes.” The report serves as the culmination of the working group and concludes, with respect to the evidence-base for the impact of stakeholder engagement (see manuscript reference 14, pg. 14):

“While previous research touted the promise of engagement, it also highlights a paucity of evidence that engagement leads to improved management of complex environmental problems.

To what extent, under what conditions, and for whom stakeholder engagement leads to changes in social and environmental conditions remains uncertain and unsubstantiated.”

Nevertheless, making a conclusive statement that *there is no gray literature that shows empirical evidence that this is true* would require us to perform a systematic review and synthesis of the gray literature, which is beyond the scope of this paper. Therefore, we edited this sentence.

- See lines 61-62: Nevertheless, there is little empirical evidence evaluating the impact of stakeholder integration on natural resource outcomes^{8,12,13}.

We are very interested in this literature, so if there is other gray literature that Reviewer #3 suggests that we review and include within our paper, please let us know.

Line 83: I think it would help to have a short description of each of the three stakeholder types - agriculture, domestic, and environment. The figure is useful but doesn't clarify exactly what the definition of these terms are. Does domestic include public supply wells? And if the environment is representative of groundwater-dependent ecosystems, what does that include/mean? I now see this is in the methods section at the end. I still think it would be helpful to add a very short description here, but fine if authors disagree.

We agree with your comment. In addition to a description in the Methods section, we have also added clarity to the main text in the following three ways:

First, we added text to Fig. 1a to explicitly define each stakeholder.

Second, we moved up our text about stakeholders in the introduction.

Third, we added clarity by adjusting the description of these stakeholders in the introduction.

- See lines 78-81: The objective of this paper is to assess if greater stakeholder integration into Sustainability Plans leads to better outcomes for those stakeholders. We focus on three stakeholder groups: agriculture, domestic, and environment (Fig. 1a). These three groups represent groups of individual users that self-supply groundwater for diverse uses, such as food production, household supply, and ecosystem function.

Line 145: Add italics to describe

Change made.

Lines 199-200: I find this a bit confusing. I think it would help to add a short description of what is meant by "inequity." Otherwise, there could be confusion between the higher proportions of each protected here when compared to the proportions of each protected at a distance of 2.4km.

We edited the whole paragraph to clarify this result; additionally, we removed the word inequity and explicitly described the trend we found in terms of disparity amongst the stakeholder groups coverage and protection.

- See lines 177-184: We ran a sensitivity analysis, adjusting the horizontal distance surrounding each monitoring well — 0.8 km to 4.0 km — to assess the impact on coverage and protection

(Supplementary Table 3.1, Supplementary Figs. 3.2-3.7). As the horizontal distance increased, the proportion of wells and ecosystems covered and protected within each Plan increased. The increases in protection did not scale proportionally across stakeholders. For example, using the largest horizontal distance (4.0 km), 78%, 77%, and 61% of agricultural wells, domestic wells, and ecosystems were covered, but only 64%, 59%, and 15% were protected, respectively. In short, the disparity in protection among agricultural, domestic, and environmental stakeholders is exacerbated as coverage increases.

Line 341: I understand why "our" is being used here, but it strikes me as an oddly possessive term for the four components of stakeholder integration that this study evaluated and measured. Suggest changing to "the."

Change made.

REVIEWERS' COMMENTS

Reviewer #2 (Remarks to the Author):

The manuscript is much improved. I appreciate the authors engagement with my comments and their extensive efforts to revise the piece. Overall, I am satisfied with the changes made although I do have a few small suggestions I offer here for consideration:

1. How were irrigation districts and reclamation districts chosen as the focused for assessing closed space representation? In my experience California water districts are more perhaps more common than reclamation districts. Clarity on this decision would help support the additional engagement analyses.
2. How do the authors understand the negative association between engage2 and coverage? To me this is an interesting finding that merits some more consideration/discussion.
3. The findings related to the lack of relationship/sequencing of the four integration components as well as the findings related to the additional analyses related to my previous suggestion of including controls highlight important policy implications that are worth lifting up. I believe this could be done without adding much to the word count.

REVIEWER COMMENTS

This document is a thorough point-by-point response to each comment made by the reviewers. **Each comment made by a reviewer is in blue-bold font.** Our responses are in black. In places where the reviewer asked for a change that led to substantial edits to the text, we provided the edited/updated text. *Changes to the text or new, additional text are noted in red italic font.*

Reviewer #2 (Remarks to the Author):

The manuscript is much improved. I appreciate the author's engagement with my comments and their extensive efforts to revise the piece. Overall, I am satisfied with the changes made although I do have a few small suggestions I offer here for consideration:

We thank Reviewer #2 for their continued engagement and feedback. We are glad that the reviewer found the manuscript improved and that they are satisfied with the changes made. We appreciate the opportunity to further refine the manuscript.

1. How were irrigation districts and reclamation districts chosen as the focus for assessing closed space representation? In my experience California water districts are perhaps more common than reclamation districts. Clarity on this decision would help support the additional engagement analyses.

Thank you for the opportunity to refine this distinction. To clarify, we did not select irrigation and reclamation districts to assess closed space representation. Rather, we found the distinction between closed versus invited spaces a useful frame to reflect on the power differential in the stakeholder groups' access to decision making spaces. Irrigation and reclamation districts are entities that provide water primarily for agricultural use, as compared to more general-purpose entities like cities or counties that manage water, land, and other domains. Although California Water Districts also typically only manage water, in most cases, they manage supply for many different uses (i.e., agricultural, domestic, industrial, municipal). California Water Districts acting as Groundwater Sustainability Agencies also represent closed decision making, but they do not necessarily represent agricultural interests. Therefore, irrigation and reclamation districts represent a special case in which primarily agricultural water users have the authority to form a Groundwater Sustainability Agency to represent agricultural water use in Sustainable Groundwater Management Act implementation. As such, we focus on irrigation and reclamation districts to represent access to closed Groundwater Sustainability Agency decision making for agricultural users.

To clarify this distinction, we revised the following language to the description of the sensitivity analysis:

- We ran a sensitivity analysis with alternative definitions of engage for agriculture — definitions which accounted for the group's power to form a Groundwater Sustainability Agency via a governing entity *that provides water primarily for agricultural use, e.g.,* a reclamation or irrigation district, which is not an option available to the other stakeholder groups (Supplementary Section 5). *Irrigation and reclamation districts are governing entities that in practice serve primarily agricultural interests and can act as Groundwater Sustainability Agencies. We anticipate that this method of accounting for agricultural representation is a conservative estimate, as it is likely that agricultural interests are represented by other entities serving as or on Groundwater Sustainability Agency boards that were not included here.*

Additionally, we added the following text, similar to the response above, to Supplementary Section 5:

- *Irrigation and reclamation districts are entities that provide water primarily for agricultural use, as compared to more general-purpose entities like cities or counties that manage water, land, and*

other domains. While other entities that manage water in California, such as California Water Districts, also represent 'closed' decision making spaces, these entities typically manage water supply for many different uses (i.e., agricultural, domestic, industrial, municipal). Therefore, we focus on irrigation and reclamation districts in our sensitivity analysis as they represent a special case in which primarily agricultural water users have access to the 'closed' space of Groundwater Sustainability Agency decision making in the Sustainable Groundwater Management Act implementation.

2. How do the authors understand the negative association between engage2 and coverage? To me this is an interesting finding that merits some more consideration/discussion.

Thank you for raising this finding. As we discuss in the paper, we see this result as indicative of a less visible form of decision making in which the structure of the representative monitoring system is decided. As we indicate in our discussion, our results suggest that the presence of irrigation and reclamation districts in Groundwater Sustainability Agencies may influence the structure of the representative monitoring system. Without further analysis, we cannot make additional claims as to the rationale for this result. Instead, we can conclude that when irrigation and reclamation district Groundwater Sustainability Agencies are present, groundwater elevation trends in agricultural wells are less likely to be represented by the representative monitoring system.

3. The findings related to the lack of relationship/sequencing of the four integration components as well as the findings related to the additional analyses related to my previous suggestion of including controls highlight important policy implications that are worth lifting up. I believe this could be done without adding much to the word count.

We appreciate that Reviewer #2 found these additional findings worthy of highlighting within the manuscript. To address these comments and stay within the journal's word count limit, we made two revisions. First, we revised a sentence in the discussion to address the reviewer's comment on the sequencing of the integration components (new text is in *red italics*):

- Our results reflect varied incentives and motivations to address stakeholder needs, *and are in some cases likely* the result of more explicit regulatory requirements (*e.g., describe for the environment as shown in Supplementary Table 2.1*), *in other cases due to local dynamics that prioritize one or more groundwater uses over others, or both.*

Second, we added a sentence to the limitations section that highlights the findings of the additional analyses on year of Sustainability Plan submission and population of wells and groundwater-dependent ecosystems:

- *From these additional analyses, we found (1) integration scores, coverage and protection were higher in the Sustainability Plans submitted in 2022, and (2) for each stakeholder group, the number of wells or acres is significantly and negatively associated with coverage, suggesting that as the number of wells and size of a Sustainability Plan area grows, the representative monitoring networks do not scale to cover the increased number of stakeholders within a Plan.*